# The Holes of Zn Phosphate and Hot Dip Galvanizing on Electrochemical Behaviors of Multi-Coatings on Steel Substrates

**Thiago Duarte** [1], **Yuri A. Meyer** [1] **and Wislei R. Osório** [1,2,*]

1    School of Technology, Campus I, University of Campinas/UNICAMP, Limeira 13484-332, SP, Brazil; gradduarte@gmail.com (T.D.); meyeryuri@gmail.com (Y.A.M.)
2    School of Applied Sciences, FCA, Research Group in Manufacturing Advanced Materials (CPMMA), Campus II, University of Campinas/UNICAMP, Limeira 13484-350, SP, Brazil
*    Correspondence: wislei1@unicamp.br

**Abstract:** The aim of this investigation is focused on the evaluation of distinctive coatings commonly applied in the automotive industry. The resulting corrosion behavior is analyzed by using electrochemical impedance spectroscopy (EIS), equivalent circuit (EC) and potentiodynamic polarization curves. The novelty concerns a comparison between tricationic phosphate (TCP), cataphoretic electrodeposition (CED) of an epoxy layer, TCP + CED and HDG (hot-dip galvanized) + TCP + CED multi-coatings. Both the naturally deposited and defect-induced damage (incision) coatings are examined. The experimental impedance parameters and corrosion current densities indicate that multi-coating system (HDG + TCP + CED layers) provides better protection. Both planar and porous electrode behaviors are responsible to predict the corrosion mechanism of the majority of samples examined. Although induced-damage samples reveal that corrosion resistances decreased up to $10\times$, when compared with no damaged samples, the same trend of the corrosion protection is maintained, i.e., TCP < CED < TCP + CED < HDG + TCP + CED. It is also found that the same trend verified by using electrochemical parameters is also observed when samples are subjected under salt spray condition (500 h). It is also found that porous electrode behavior is not a deleterious aspect to corrosion resistance. It is more intimately associated with initial thickness coating, while corrosion resistance is associated with adhesion of the CED layer on TCP coating. The results of relative cost-to-efficiency to relative coating density ratios are associated with fact that a CED coating is necessary to top and clear coating applications and the TCP + CED and the HDG/TCP + CED coating systems exhibit the best results.

**Keywords:** electrodeposition coatings; automotive; corrosion resistance; EIS; multicoatings

## 1. Introduction

Corrosion is a natural material degradation in contact with the environment and its oxidative elements. In general, corrosion spontaneously happens because these materials have higher Gibbs energy than their fundamental state [1]. In the case of steel, it corrodes to many oxidated species, such as magnetite ($Fe_3O_4$) and iron oxide (FeO). These species have a composition that close to the raw material that originates this alloy [2,3]. For this reason, the corrosion maintenance costs to maintain metallic structures and consume goods are relevant [2]. In a study conducted by Technologies Laboratories, Inc., Federal Highway Administration (FHWA) and National Association of Corrosion Engineers (NACE) International, the estimated annual cost with corrosion is around $276 billion, and 8% of this value is related to the automotive sector including manufacture processes, repairs and maintenance, and depreciation [4]. In general, the car bodies are combined with distinct substrates, each one of which has an engineering application. From these, cold-rolled steel (CRS), higher-strength steel, aluminum alloys, galvanized steel, and others are mentioned.

In this context, the steel substrate stands out due to its versatility in the alloy manufacturing, its mechanical properties, and its cost-benefit, representing 60–70% of the final weight of the car. This constitutes a great challenge since the steel substrate is more susceptible to corrosion than galvannealed or aluminum substrates [4]. By these factors, the automotive industry has improved the surface treatment to minimize the steel corrosion effects. The pretreatment processes with tricationic phosphate (TCP) and the painting process with cathodic electrodeposition (CED) were developed in the 1940's and 1970's, respectively. These mentioned processes increase the metallic substrate corrosive protection following the environmental regulatory exigencies and manufacture financial expectations [5–9].

TCP is a protective layer that is chemically constituted to form a coating on the metal surface to protect against corrosion. It is one of the most commonly used metal protective coatings, and has an easy operation, good adhesion and wear properties [10–12]. Although the Zn phosphate conversion is more widely used in the automotive industry than other types of phosphate coatings [10–12], only its corrosion resistance is not yet satisfactory due its constituent porous structure [10–14]. It has been reported that a CED has played an important role in improving the corrosion protection of cars parts [15–17]. Usually, in the automotive painting process, after the TCP layer, a deposition of a CED layer is carried out. This layer is commonly constituted of anticorrosive pigments and epoxy resin. Due to the immersion paint process, the CED guarantees a uniform paint layer in the internal and external parts of the car body, and corrosive protection on the edge substrate [4,13–15].

In addition to these surface treatments, hot dip galvanized (HDG) steel sheets are widely used in building and automotive applications [18]. This commonly constitutes a coating type with zinc containing a small amount of aluminum to suppress Zn-Fe layer formation during galvanizing steel from pure molten zinc. It has also been reported that other types of Zn coatings are alloyed with higher amounts of aluminum and aluminum and magnesium in order to improve the corrosion resistance of the Zn coatings [18].

The corrosion mechanism of a coating generally involving diffusion of corrosive species though coating and their transportation at interfaces is substantially important to predict the lifetime of coating or multi-coatings systems [19]. It is recognized that there is a demand to predict or describe the corrosion performance of substrates. For this purpose, the corrosion tests commonly take place over long-term period of time. In this direction, salt spray techniques (ASTM B117, ~40 days), cyclic corrosion tests (VDA 621–415, ~70 days) and VICT 1027 (6 weeks) [4,5,16] are typically carried out. In this scenario, electrochemical impedance spectroscopy (EIS) and potentiodynamic polarization have been used with great potential as alternative to accelerated corrosion tests for evaluating metallic-coated surfaces [14–18].

It is known that distinctive conversion coatings (e.g., chromate, phosphate, molybdate, zirconium, cerium) used to improve the epoxy (organic) coating adhesion and corrosion resistance have been investigated [20–22]. Particularly, those containing chromate [20] are toxic and/or carcinogenic, which have the tendency to been banned. From these mentioned coatings, the phosphate coatings, in particular Zn phosphate (tricationic) conversion, is one that is widely used in the automobile industry and in the substrate covering industry [10]. Since the corrosion resistance of the phosphate is not enough due to porous its structures [10], some modified compositions have widely been investigated [10–12,20,21].

Su and Lin [23] have investigated the different effects of distinct additives on the properties of Zn phosphate coatings on a steel substrate. They have reported the positive effects of both Ni and Mn ions in decreasing the porosity level. Vakili et al. [20] have demonstrated that both adhesion and corrosion resistance are improved when Zn phosphate modified with Ce and Zn is used [20].

In the automotive industry, a system of multi-coating layers constitutes the final coating. It is initiated with an inorganic Zn phosphate layer (e.g., TCP) followed by an electrodeposited coating (e.g., CED), a primer layer to improve mechanical property, and finally, a polymer coating (e.g., acrylic, epoxy, based-coat/clear-coat layers) constituting the aesthetic aspect [24]. It is reported that each one of these layers has a distinct role to play

in the overall performance of the coating system [24]. It is recognized that the diffusion of corrosion species throughout polymer coating and ions that are transported along the coating system, reaching the coating–metal interface, have important roles on the resulting corrosion resistance [19,20,24].

Since the 1990′s, Acamovic et al. [25] have reported that cathodic electrocoat layers directly deposited on bare steel, or on phosphated steel, are porous and prevalent and their effects upon the resulting corrosion resistances are substantial. Reichinger et al. [19] have reported that due to the permeability of water and solvated ions into coating and on Zn (when HDG substrates are considered), and during film formation, the substrate dissolves under the high alkaline condition provided. Ramezanzadeh et al. [24] have reported that hydrolytic and photo degradations affect the coating performance. Consequently, the diffusion of corrosive fluid penetrates easily and the coating–metal interface is reached. They have also demonstrated [23] that the cross-link density of the polymer coating is negatively affected, and holes (micro- and nano-sized) are formed. Thus, four main stages predict the coating performance [23]. Firstly, corrosive media penetrates through the holes. Water contained in media induces a higher level of hydrophilicity and the holes increase. Secondly, the coating–metal interface is reached. Sequentially, when an electrolyte reaches this interface, the corrosion of steel substrates is initiated (3rd stage). Furthermore, according to Ramezanzadeh et al. [24], at the 4th stage, the intermediate absorbed species or corrosion by-products eventually formed at the coating–metal interface tend to block the holes or pores. Reichinger et al. [19] and Vakili et al. [20] have reported similar observations. These have also detailed that hydroxyl ions increases local pH and that small points of delamination are provoked, which also induces adhesion decreases.

Concerning electrochemical impedance measurements, no porous electrode behavior is reported and/or discussed. When CED is applied, constituting coating system, Nyquist and Bode-phase plots reveal capacitive planar behavior (angles close to 90 degrees) [19,20]. On the other hand, when Tian et al. [10] recently investigated modified Zn phosphate on carbon steel substrates, the porous electrode behavior occurred, but was not discussed or commented upon. They demonstrated that incorporation of Zr to phosphate provides a decrease in the phosphate crystal particle size. Consequently, the resulting pores are filled (decreasing the diffusion of corrosive ions) and more dense and compact phosphate coating is attained [10]. When Acamovic et al. [25] have investigated Nyquist plots of coated bare steels after 18 days in 3% NaCl solution, with two distinctive thicknesses of CED (i.e., 10 μm and 18 μm), porous and planar electrode effects coexist. However, this was not discussed or mentioned.

It is worth noting the important role of a pretreatment on the substrate and subsequent layers constituting a coating system that is more complex. From the electrochemical point of view associated with economical and environmentally friendly aspects, it is very important to understand the ratio between thickness (or weight per area) and corrosion resistance response with the final manufacturing cost. On the one hand, it seems that a thicker coating layer induces an increased time to swell coating (electrolyte uptake) and effective degradation is retarded. On the other hand, the weight–performance ratio can attain a deleterious level.

In this proposed investigation, the aim is focused on the evaluation of four distinctive coating systems on steel substrate: TCP, CED, TCP + CED and HDG + TCP + CED coatings. The experimental results demonstrate the electrochemical responses of these distinct coating systems considering typical deposited coatings, and when a failure-induced defect (incision) is provided. The results also elucidate the role of TCP in a single coating and when a multi-coating system is considered. The EIS parameters and potentiodynamic polarization curves are analyzed and discussed. Moreover, in order to confirm the corrosion mechanism, diagrams used to verify time constants are used; and an equivalent circuit containing Warburg elements is also proposed. It is remarked that electrochemical investigations were carried out, taking initial immersion times (short term) into stagnant and naturally

aerated 0.5 M NaCl at an environmental temperature (~25 °C), and the same trends along longer-term periods are suggested.

## 2. Materials and Methods

### 2.1. Metallic Substrate and Surface Treatments

A cold-rolled steel (CRS), belonging to the group of low carbon steel [25–27], was used as a substrate. The CRS substrate samples were withdrawn in dimensions 30 ($\pm$1.5) cm $\times$ 10 ($\pm$1) cm. The CRS samples containing tricationic phosphate (TCP) conversion coatings were also used. These samples were acquired from a supplier ACT Test Panels LLC Company, Hillsdale, MI, USA (https://acttestpanels.com/, accessed on 19 March 2022). Table 1 shows the CRS substrate composition according to ASTM 1008 and these were also certified by the supplier. A conventional phosphate conversion is considered, i.e., alkaline degreaser, titanium activation, phosphate bath and zirconium base passivation [4] with products supplied by PPG Industries, Pittsburgh, PA, USA (https://www.ppg.com/ accessed on 19 March 2022). The TCP layer had a coating weight (CW) of about CRS = 2.21 ($\pm$0.6) g/m$^2$ and HDG = 3.42 ($\pm$0.3) g/m$^2$ and a crystal size (CS) of approximately of CRS = 3.5 ($\pm$1.8) μm and HDG = 4.8 ($\pm$1.4) μm. This condition was parametrized. For CED samples, a PPG degreaser based on potassium hydroxide and surfactants were utilized in order that protective oil on the CRS could be removed. Immediately after this process, on the samples, an electrocoat paint job was carried out. For this purpose, a volume of 8 ($\pm$0.1) liters of PPG Industries bath paint lead-free composed of a water-based resin epoxy and a pigment paste was used [26]. Their corresponding physical–chemical parameters are described in Table 1. The reproducibility of the aforementioned procedure was guaranteed by using triplicate experimentation. In order to apply electrocoat paint, an Ametek$^\circledR$ DC programmable power model XG 1500 (AMETEK, Inc, Devon-Berwyn, PA, USA) was used. These panels were coated under stirring, at 30.5 ($\pm$ 0.5) °C, and the voltages and application times are shown in Table 2. These parameters were adopted in accordance with previous expertise as provided by a supplier. For the curing process, an electric furnace to reach the metal temperature at 177 ($\pm$5) °C for 30 min was used. The average thickness for the HDG + CED was 18.7 ($\pm$1.2) μm, while the HDG + TCP samples attained about 17.3 ($\pm$0.9) μm and the TCP sample of about 1.2 ($\pm$0.4) μm. Triplicate was also adopted in order to guarantee the reproducibility of the aforementioned procedure stages. Average values are considered and reported upon.

**Table 1.** Average chemical composition of the CRS substrate.

| Element | wt.% |
|---|---|
| C | 0.10 |
| Mn | 0.60 |
| P | 0.04 |
| Cu | 0.20 |
| S | 0.035 (<than) |
| Fe | Balance |

**Table 2.** The utilized CED parameters deposition.

| Substrate Samples | Voltage (V) | Time (s) |
|---|---|---|
| TCP | 210 | 120 |
| CED | 180 | 120 |

### 2.2. Panel Standardization for EIS and Salt Spray Measurements

A schematic representation used in order to demonstrate the panels where samples are to be subjected in the EIS measurements were withdrawn, as shown in Figure 1. Each sample has the dimensions of 30 ($\pm$1.5) mm $\times$ 20 ($\pm$1.5) mm and 5 ($\pm$0.5) mm of bare metal (grounded, SiC paper and polished, 0.5 μm). All the samples have their edges

protected with a solvent-born acrylic resin paint (from PPG Industries, Pittsburgh, PA, USA) with a high corrosion resistance. One part of the sample has two incisions of 0.5 mm of width, with 2 mm of distance between them. The other part did not have any incisions. The two parts remaining were used to perform the salt spray measurements. These were conducted according to the ASTM B 117 standard using a chamber (Model SST-B, Ten Billion Co., Taiwan, China) under an atmosphere containing 5 wt.% NaCl solution, pH of 7, and conditions were room temperature. The samples are displaced in a perpendicular direction, forming 30° of inclination. These were kept under this condition, receiving salt spray continuously for 500 h.

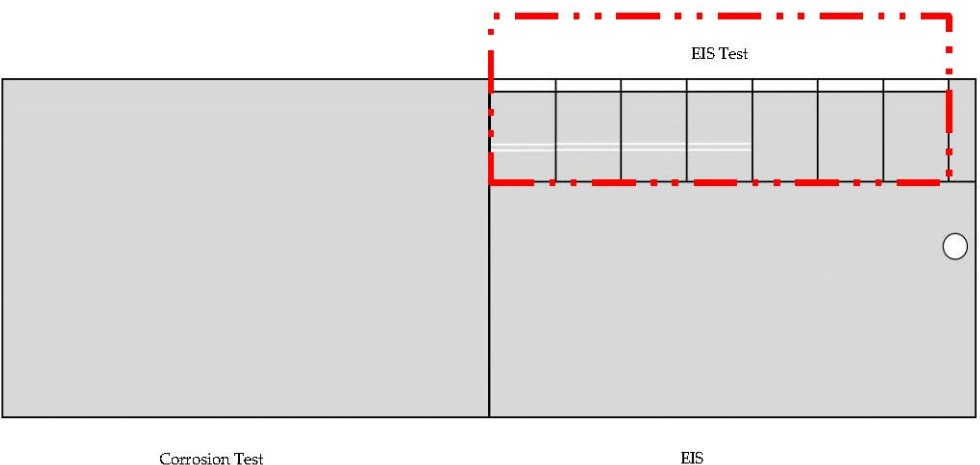

**Figure 1.** A schematic representation of the panel displacing the area where the samples for EIS were withdrawn.

*2.3. EIS and Potentiodynamic Polarization Measurements*

In order to carry out the electrochemical measurements, a conventional three-electrode cell kit and an Ametek® Potentiostat Galvanostat VersaSTAT 4 (AMETEK, Inc., Devon-Berwyn, PA, USA) was used. A platinum plate (1000 ($\pm$10) mm$^2$) as a counter electrode and working electrodes (WE) as coated samples with an examined area of about 100 ($\pm$5) mm$^2$ were used. A saturated calomel electrode (SCE) as a reference electrode was also utilized. For each measurement, this set is immersed into a volume of 80 ($\pm$2) mL of a stagnant and naturally aerated solution of 0.5 M NaCl at an environmental temperature of 25 ($\pm$5) °C. Before starting the electrochemical readings, all samples were kept for 10 min and were immersed into the mentioned electrolyte in order for a steady-state to be attained. This is a practice that is commonly adopted and is intended to stabilize distortions and oscillations [27–29]. During this process, open-circuit potential (OCP) curves were observed, and no variations greater than 5 mV per second were observed.

EIS experimentations were carried out (~40 ($\pm$1) min.) before the potentiodynamic polarization techniques were carried out (during of about 1 h). The EIS measurements were carried out by using a potential amplitude of 10 mV, peak-to-peak (AC signal) in open-circuit with 10 points per decade. A frequency range between $10^5$ Hz and $10^{-2}$ Hz was adopted [27–29]. In order to determine the impedance parameters using simulations correlated with the experimental data obtained, the complex non-linear least squares (CNLS) simulations were carried out [27,29]. A ZView® software (version 2.1b), Scribner Ass. Inc., Southern Pines, NC, USA, was used to propose an equivalent circuit and obtained the data. These were organized and plotted by using Microcal Origin® (version, XX), OriginLab Corporation, Northampton, MA, USA.

Concerning the potentiodynamic polarization measurements, a scan rate of about 0.167 mV/s scanning from −300 mV to −850 mV (vs. SCE) for all of the examined samples were adopted, except for those samples with pretreatment of HDG steel samples, which ranged between −550 mV to −1300 mV (vs. SCE). A Tafel extrapolation method was adopted to obtain the corrosion current densities ($i_{corr}$). Based on the cathodic and anodic

branches, the averages of corrosion potential ($E_{corr}$) and $i_{corr}$ values were considered. In order to guarantee the reproducibility, a duplicate was considered.

### 2.4. Coated Surface and Incision Characterizations

After EIS measurements were carried out, the coated surface and the incisions were characterized by using a scanning electron microscope (SEM), model VEGA3 from TESCAN®, Brno, Czech Republic, plated with an energy-dispersive x-ray (EDX) detector. Thus, the profile of the incision and the presence of a corrosion by-product was characterized. Additionally, in order to evaluate the phases constituted in all examined samples, a PANalytical® XPert diffractometer (X'Pert model), Malvern, Worcestershire, UK, operated under 40 kV and 30 mA with Cu K$\alpha$ radiation and a wavelength of 0.15406 nm was used.

## 3. Results and Discussion

### 3.1. Potentiodynamic Polarization Curves

Figure 2a,b shows the obtained experimental potentiodynamic polarization curves of the all examined samples without incision (or damage at surface) and contained damage, respectively. These curves are representative from a duplicate that was carried out. The examined samples are designated as TCP, CED and TCP + CED considering the sample both with and without incisions provoked. From those results without an incision, it is evidenced that the lowest corrosion current density ($i_{corr}$) is that of the TCP + CED sample, attaining about 0.012 ($\pm$0.002) $\mu Acm^{-2}$, with a corrosion potential ($E_{corr}$) of $-429$ mV (vs. SCE). The intermediate $i_{corr}$ is that of the CED coating and the corresponding highest $i_{corr}$ value is that of the TCP sample (i.e., 1.59 ($\pm$0.15) $\mu Acm^{-2}$). This indicates that by only using a TCP coating system with a thickness varying between 1.5 and 2.5 $\mu$m, an increasing in the corrosion current density, higher than 100 times that in the $i_{corr}$, is attained when compared with the TCP + CED sample. When the TCP sample is compared with the CED sample (thickness between 5 $\mu$m and 7 $\mu$m), still an increase in the $i_{corr}$ greater than 30 times is recorded. This suggests that by only using the TCP coating system on a CRS steel substrate is the worst corrosion behavior provided.

As expected, when the incisions are provoked at various surfaces of the coated samples, all of the examined samples that demonstrated $i_{corr}$ values were substantially increased. This seems to be correlated with the higher active anodic areas exposed rather than no damaged surfaces. These experimental results indicate that the TCP has increased ~1.6 $\mu Acm^{-2}$ to 10.2 ($\pm$0.1) $\mu Acm^{-2}$, i.e., by about 6$\times$. The CED sample has also increased its $i_{corr}$ of ~0.047 to 3.7 ($\pm$0.2) $\mu Acm^{-2}$ (~80$\times$), while the TCP + CED coating system with an incision has its $i_{corr}$ increased by about 160$\times$, i.e., of 0.012 to ~2 $\mu Acm^{-2}$. Considering the $E_{corr}$ values, it can be said that variations of up to 60 mV (vs. SCE) are observed. However, no clarified conclusions concerning the displacement to nobler or less nobler side potentials can be attained. It was also found that the cathodic plateaus also slightly (~100 mV (vs. SCE)) increased when the incisions were constituted. It is also worth noting that oscillations observed in curves, mainly in cathodic branches, are more prominent in curve damage-induced samples (with incisions). The oscillations seem to be more associated with the nature of the substrate and with electrolyte penetration into coating systems, forming possible galvanic couples. Potentiodynamic polarization curves due to low carbon steels in a 0.5 M NaCl solution have been reported, and no typical oscillations were verified [27,28].

Based on the experimental polarization curves, it can be concluded that, independent of the incision or no damaged surfaces, the same corrosion resistance trend is verified, i.e., TCP < CED < TCP +CED. Evidently, more systematic investigations concerning corrosion resistance should be carried out in order to predict the electrochemical behavior of these samples examined. In this sense, EIS measurements were carried out and are discussed in the next section. Additionally, the thickness of each examined coating system seems to help understand the electrochemical behavior.

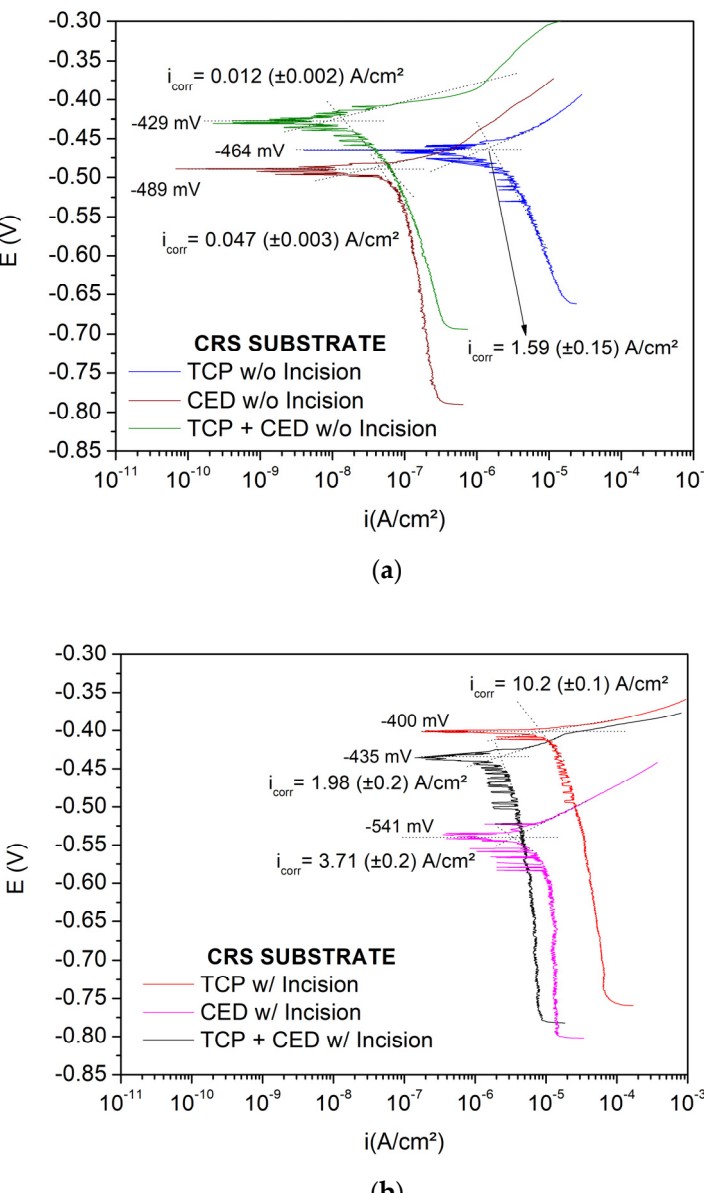

**Figure 2.** Experimental potentiodynamic polarization curves of three distinct coating systems (TCP, CED and TCP +CED) in an 0.5 M NaCl solution, platinum plate as counter electrode and SCE as a reference electrode, where: (**a**) the samples have no incisions (damage-induced samples), and (**b**) the samples with incisions expose the steel substrate.

## 3.2. Number of Time Constants

The number of time constants can be intimately associated with distinct reactions affecting the electrochemical behavior of the samples examined. An understanding of these numbers of time constants is useful in describing the mechanism of corrosion, intermediate absorbed species and the film of corrosion by-products that constitute a protective barrier, providing transport and diffusion to these species. Thus, before discussing EIS plots, the number of time constants were analyzed. For this purpose, the moduli of imaginary parts of the impedances per frequency (in logarithm) are plotted, as shown in Figure 3.

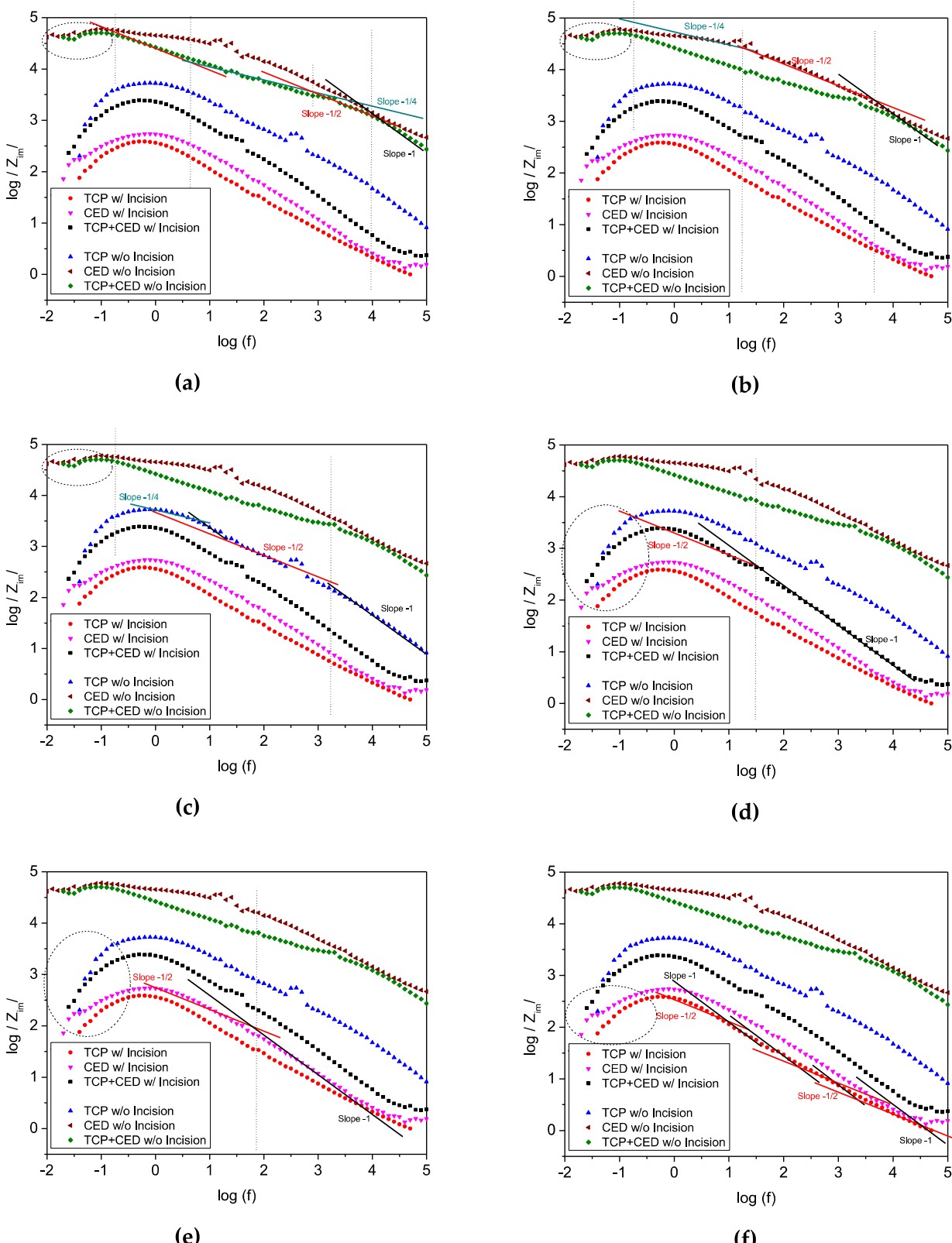

**Figure 3.** Moduli of the imaginary parts of the impedances with frequencies of the coating system samples without incisions (damages induced): (**a**) TCP + CED sample, (**b**) only CED coating and (**c**) TCP coating; and when incisions are provoked: (**d**) TCP + CED, (**e**) CED and (**f**) TCP coating systems, respectively. These results are obtained when a 0.5 M NaCl solution at environmental temperature is considered.



This is a method that was previously reported by Hirschorn, Orazem, and Tribollet [30,31], and was also reported in other studies [29,32–34]. Figure 3a–c show the results of the modulus of the imaginary part of the impedance, with a frequency of the TCP + CED, CED and only the TCP sample, without incisions provoked. Figure 3d–f depict those same samples containing the incisions or damages induced, respectively. Considering the samples of the coating systems without the incisions (Figure 3a–c), in a general way, the slopes occur at $-1$, $-1/2$ and $-1/4$ at frequency domains that are high ($>10^3$ Hz), medium (between 10 Hz and $10^3$ Hz), and low ($<1$ Hz), respectively. Thus, it can be said that three constant times are prevalent. A first time, corresponding more with the external coating layer in direct contact with electrolyte; a second time, associated with reactions at the interface between the last coating interface with the substrate (forming intermediate species); and a third time, which is constituted between the corrosion by-product into the steel substrate and at the interface of corrosion by-products, the substrate and the penetrated electrolyte (transport and diffusional phenomena throughout a porous structure layer). Those slopes characterized at $-1/2$ characterize the participation of a porous electrode behavior in the corrosion process [29–36]. It has been reported that a porous electrode behavior can also be characterized at slopes of $-1/2$ and $-1/4$ [29,35,36]. When Nyquist plots are discussed, these slopes constitute straight lines, forming $45°$ and $22.5°$, which are demonstrated and discussed [35,36].

When the coating system samples containing incisions are analyzed, only two time constants are prevalent, as shown in Figure 3d–f. This seems to be associated with the exposed CRS substrate (steel bare), independent of the nature of the coating considered. Although only two constant times are observed, the porous electrode behavior is still characterized. This will be explained when EIS parameters are forwardly discussed. These two constant times, appearing in damage-induced samples, seem to be associated with the fact that the coating system has no prior protective barrier. Thus, a first constant time is correlated to the electrolyte and prior coating layer, and the second with the substrate/coating interface. This is better comprehended when Bode-phase and Nyquist plots are analyzed and discussed in the next section.

### 3.3. EIS Measurements: The Effect of Incision and Porous Electrode Behavior

It is verified that three possible constant times prescribe the corrosion behavior of the examined samples without induced-damage. Two constant times help to understand the corrosion mechanism of those damaged samples (with incision). Based on this, a qualitative analysis of the Bode diagrams of those samples without (w/o) incisions and the samples with an incision (w/ Incision) provide important information, as shown in Figure 4a,b, respectively. A first analysis shows that the TCP + CED and CED coating systems, which have three constant times, have distinctive phase angles at a high frequency domain when the induced-damage samples are compared. Both the TCP + CED and the CED coatings have a phase angle higher than $60°$, while the TCP sample has a phase angle closer to zero. This is intimately associated with the porous characteristic of a typical TCP coating system [10–12,20,21,23–25].

It is important to differentiate between a porous material as constituted when a TCP layer is formed, and a porous electrode behavior. A porous structured material, when subjected or immersed into a corrosive medium, electrochemically provides a porous electrode behavior. Depending on the involved intermediate species and its evolution (transport and penetration/diffusion), a barrier protective oxide film can be formed, and corrosion is substantially reduced. On the other hand, when ions are transported and diffusion occurs, the corrosion is reinitiated, and a drastic and severe degradation takes control of the domain of the corrosion mechanism.

When the samples with and without incisions are compared, it is evidenced that, at the low frequency domain, the experimental moduli of impedances are decreased. Moreover, the kinetic of the double layer formation is slightly different to those samples without incisions (two and three constant times, as previously determined). Besides, Bode-phase

plots reveal that all examined samples with incisions have at high frequency domain, and their corresponding phase angles are close to zero.

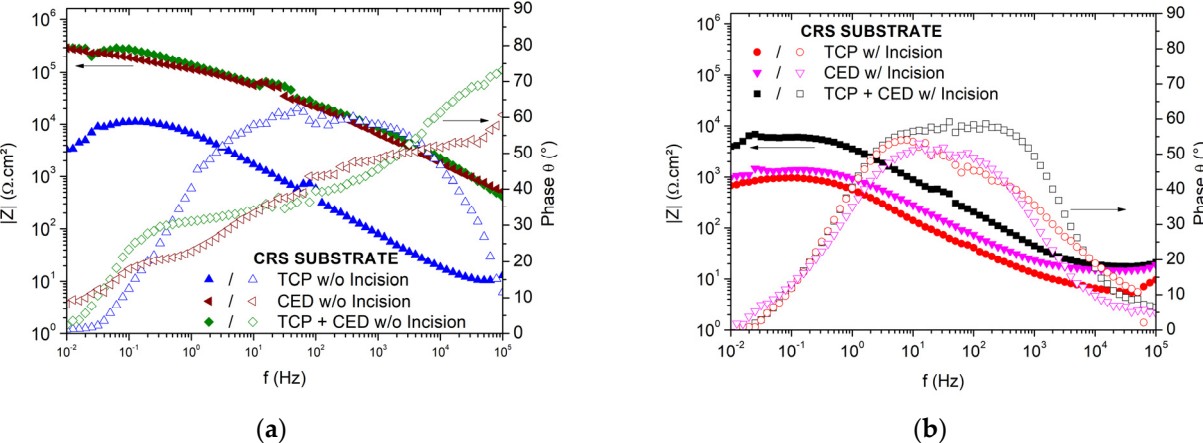

**Figure 4.** Experimental results of EIS in Bode and Bode-phase representations of the TCP, CED and TCP + CED coating systems in 0.5 M NaCl with: (**a**) without incision and (**b**) with incision (damage exposing the CRS substrate).

This is considerably distinctive when the CED and TCP + CED coating systems are analyzed. Similar observations were reported [10,19,20] concerning the higher phase angle (at high frequency), favoring a sample with a preserved coating layer, and the substrate (more active material) is not immediately exposed to an electrolyte.

Based on these Bode and Bode-phase plots, considering a qualitative analysis, the same trend is verified in the potentiodynamic polarization curves, i.e., the worst corrosion behavior is that of the TCP, followed by only CED coating and finally, the best corrosion resistance is that of the sample containing the multi-coatings applied (CED on TCP and this on CRS substrate). Although these EIS plots have demonstrated the same corrosion tendency, its new contribution concerns the number of the constant time, which helps to predict the reactions that occur and the kinetic of the double layer formation. However, some other complementary information that is not provided can be achieved when Nyquist plots are evaluated/analyzed. In this sense, the resulting Nyquist plots of the examined TCP, CED, and TCP + CED coating systems without an incision are show in Figure 5.

Figure 5a depicts the depressed semi arcs with corresponding CNLS simulations, as will be discussed further on. The same results of Nyquist plots in the lower scale range of both Z imaginary ($Z_{Im}$) and Z real ($Z_{Re}$) are shown in Figure 5b. With this, the substantial difference in the magnitude of the semi arcs corresponding with the TCP + CED and CED samples and compared with the lowest semi arc of the TCP coating sample, can be observed. Additionally, at a high frequency, the domain ranges up to about 10 Hz, and the straight lines forming 45° are characterized. Between 10 Hz and ~0.25 Hz, the TCP + CED samples trend to form 22.5°, which seems to correspond with transport and diffusional phenomena. Above ~0.25 Hz to the lower frequency domain range, a distorted capacitive arc seems to be associated with finite or semi-infinite pores and with Warburg behavior, as reported by Macdonald [37], Bastidas [35] and Murray [36].

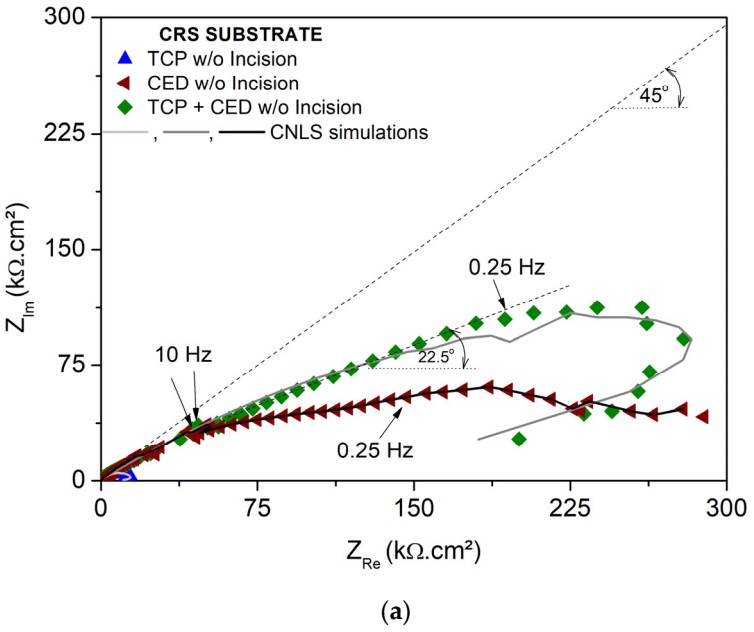

(**a**)

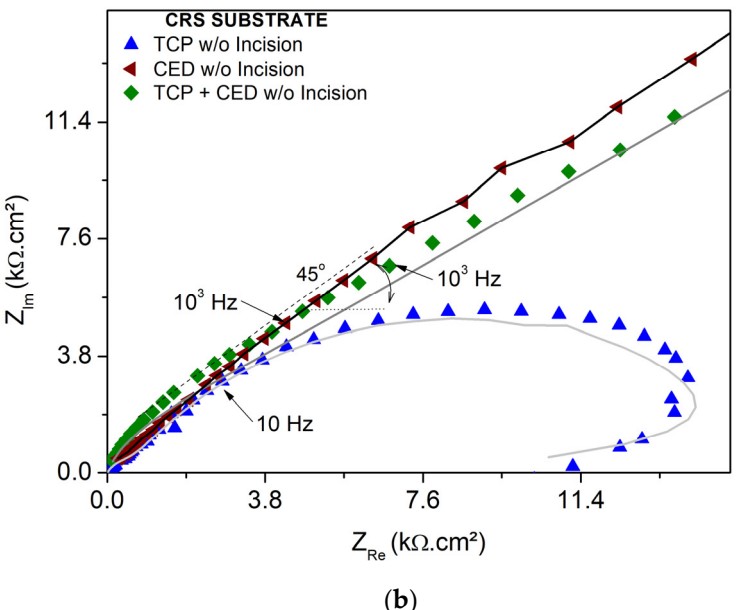

(**b**)

**Figure 5.** Nyquist plots of the TCP, CED and TCP + CED coating systems in a 0.5 M NaCl solution without incision depicted in two distinct scale ranges: (**a**) evidencing the completed depressed semi arcs of the CED and TCP + CED coating system samples and (**b**) detailing the lowest semi arc of the TCP samples. Straight line forming 45° characterizes porous electrode behavior.

With these observations, it was induced that both the planar and porous electrode behaviors are responsible for predicting the corrosion resistances of all the samples examined. It is remembered that a porous electrode behavior is described by Equation (1):

$$Z = (R_0 Z_0)^{1/2} \, coth \, (L \sqrt{(R_0/Z_0)}) \tag{1}$$

where $R_0$ and $Z_0$ are the electrolyte resistance ($\Omega$/cm) and the interfacial impedance for one-unit length, and L means the length of each pore in cm [35–37]. When the angle with the $Z_{Re}$ tends to 45°, L tends to a semi-infinite condition, and the *coth* term reaches 1. This indicates that Warburg impedance is half of that reached by a flat electrode [35–37]. This can

also be associated with those verified slopes of −1/2, −1/4 and −1/8, when the modulus of imaginary impedance parts with frequency was shown in Figure 2 (constant of time). Figure 6 shows the depressed semi arcs and CNLS simulations of the TCP + CED, CED and TCP coating samples with the damage-induced incision. With this, it is possible to compare qualitatively the samples without incision and with incision.

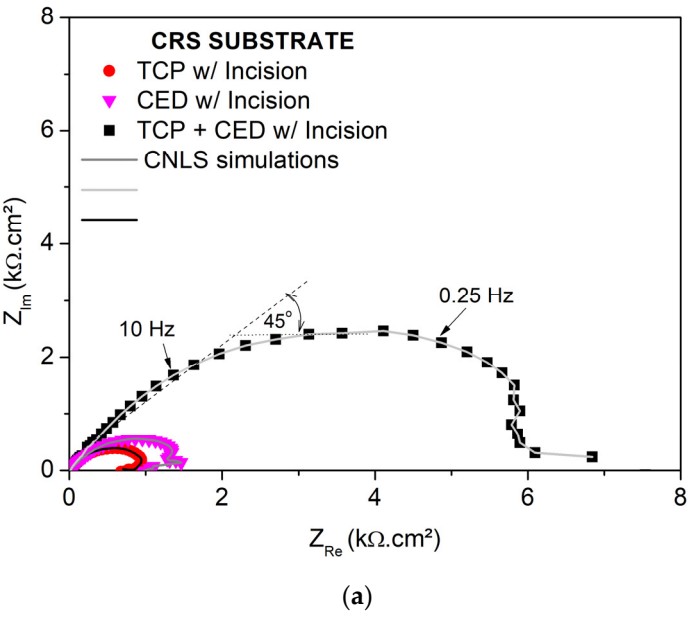

(**a**)

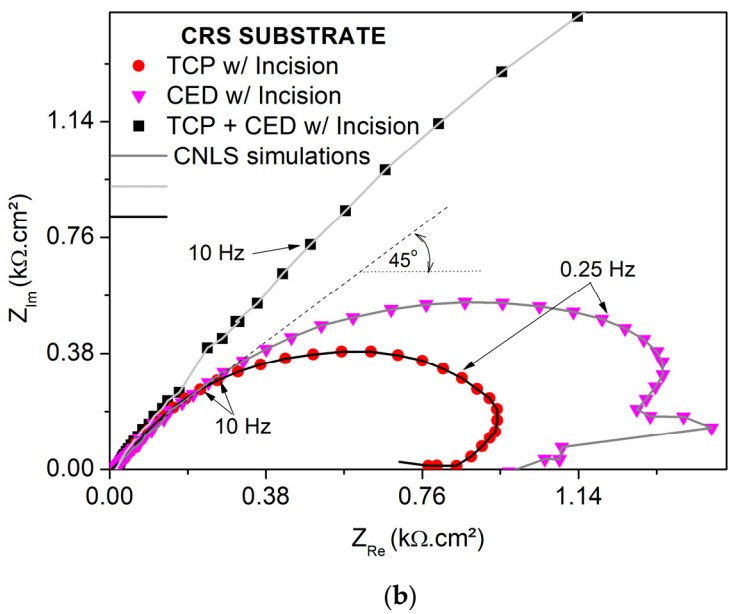

(**b**)

**Figure 6.** Nyquist plots of the TCP, CED and TCP + CED coating systems in a 0.5 M NaCl solution with incision depicted in two distinct scale ranges: (**a**) evidencing the completed depressed semi arcs and (**b**) detailing semi arcs in a low frequency domain, characterizing the straight line of 45° (porous electrode behavior).

This provides the assertion that the incision has provoked a substantial decrease in those verified semi arcs of all the examined samples. However, the same trend is verified when the polarization curves and EIS plots were analyzed, and it is also observed for these damaged samples, i.e., the increasing tendency of the corrosion resistance is TCP + CED

> CED > TCP in the coating sample. Two distinctive magnitudes of the $Z_{Im}$ vs. $Z_{Re}$ are shown in Figure 6a,b, i.e., up to 8 and 1.5 k$\Omega \cdot$cm$^2$, respectively.

In order to carry out a quantitative analysis concerning EIS measurements, CNLS simulations using an equivalent circuit (EC) into ZView® software (previously detailed) were carried out. This adopted EC is commonly used to prescribe the corrosion behavior of distinct materials and conditions [27–30,34,35]. In this present investigation, although distinctive coating systems were analyzed, a unique EC is proposed.

In a general way, all coating systems have a resistance conjugated with a capacitance consta phase elment (CPE), and this is associated with an outer layer (external) interfacing with an electrolyte (NaCl is possibly penetrating). Another resistance and capacitance are associated with the reaction at the substrate/inner layer coating interface. Additionally, a Warburg component in series with resistance against the substrate/inner layer interface is also attributed. This corresponds with diffusional reaction and transport occurring at the interface. The proposed EC adopted in this present investigation to prescribe the electrochemical behavior of coatings has been widely reported in the literature [38–41].

Figure 7a,b shows the schematic representations when only the TCP layer and the conjugated TCP + CED coating systems were constituted, respectively. Associated with the schema of the constituent elements into the coating system involving an electrolyte and the substrate, the proposed EC is also depicted. Its complete configuration is also depicted in Figure 7c.

Amirudin and Thierry [42] and colleagues [18] have proposed a review paper explaining the corrosion mechanism of the phosphate zinc layer on steel substrate. With great scientific merit, they have explained the effects of various factors influencing the corrosion mechanism. These mentioned works help to understand the various stages observed in this investigation. Although they are not represented in Figure 7, where only CED coating is considered, it can schematically be represented by Figure 7a. In addition, Figure 7 shows the presence of the FeOOH as a corrosion by-product occurring in a CRS. Morcillo et al. [3] have detailed the importance of the FeOOH and their species in steel corrosion. Furthermore, the presence of FeOOH was found in CRS samples that were available after EIS in the present work, as will be discussed in the next section.

It is evident that the thickness is modified (~5 to 7 times higher than TCP ones) and that the porosity level is different due to the nature of a CED when compared with TCP coating. This is better discussed later on. In Table 3, the impedance parameters were obtained using a CNLS simulation, ZView®; EC and experimental data are immersed into a 0.5 M NaCl solution, and no incisions were provoked on the surfaces of the samples examined.

When these parameters were analyzed, and mainly those corresponding with $R_2$ and W (corresponding with resistance of Warburg component, $R_W$), it was clearly observed that the TCP + CED coating had the highest resistance, followed by intermediate values corresponding with those of the CED coating and the lowest was that of the TCP coating, as was also previously observed when polarization ($i_{corr}$) and qualitative EIS analyses were carried out. It was also remarked that the resistance $R_2$ was higher than all of the other examined resistances. It is also remembered that each resistance is associated with a capacitance CPE.

From the experimental results, all capacitances $Z_{CPE\,2}$ are higher than $Z_{CPE\,1}$ for all of the examined samples. However, this has not an assertive correlation. A decay trend was observed when the TCP, CED and TCP + CED coating systems were compared. This seems to be attributed with the distinctive nature of the coating systems examined. It seemed to be confirmed when both the $Z_{CPE\,1}$ and $Z_{CPE\,2}$ were analyzed.

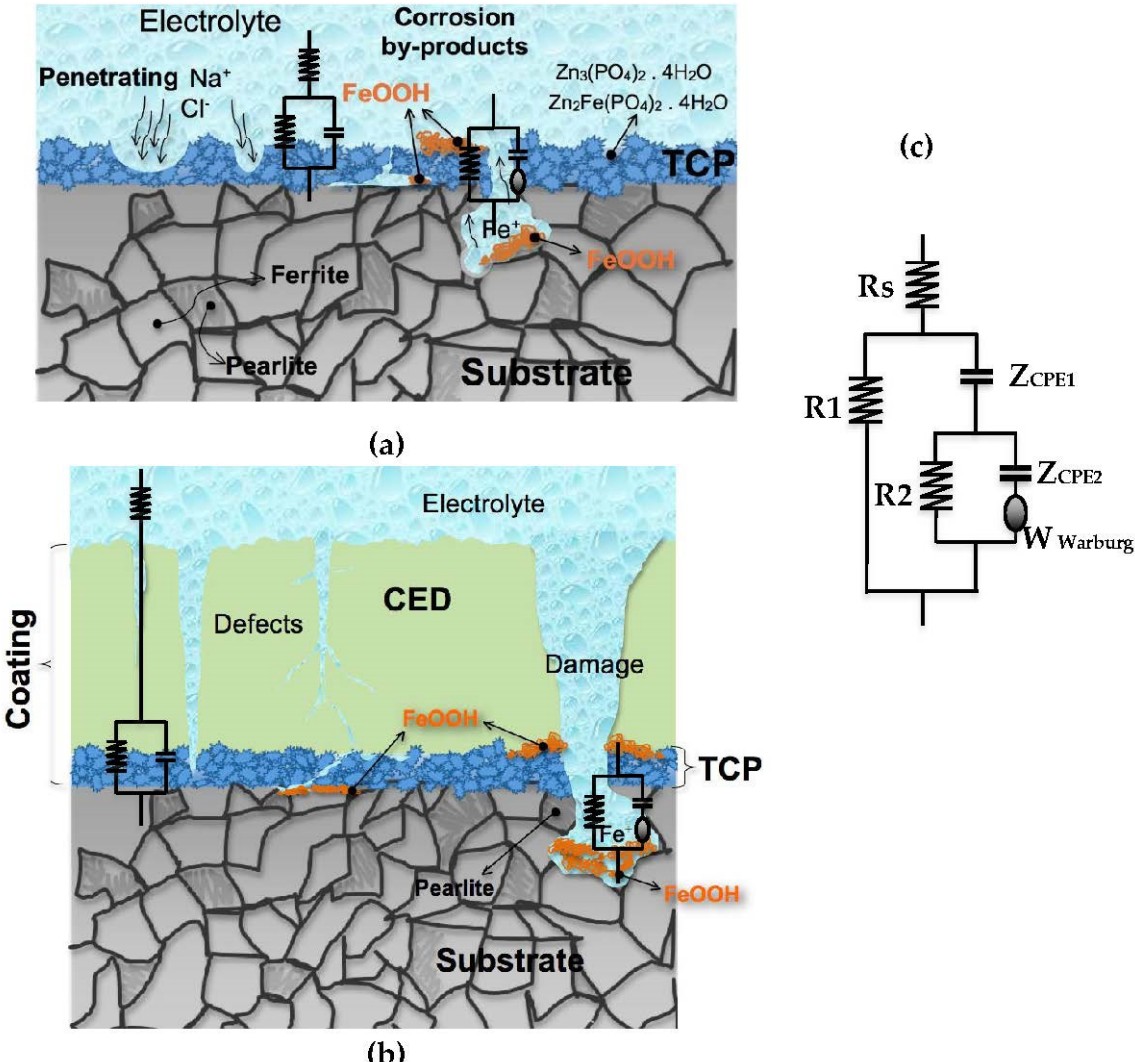

**Figure 7.** Schematic representation with an equivalent circuit proposed: (**a**) substrate and TCP coating; (**b**) substrate TCP + CED coating systems; and (**c**) complete/reorganized equivalent circuit. The resulting microstructures and microconstituents of a mild steel (similar to CRS) is agreed as in the previously reported investigation [27].

It is stated that the lowest thickness of coating is that of the TCP coating (between 1 and 1.5 μm), and the highest is that of the TCP + CED coating system (~16 μm). Associated with these values, the capacitances are inversely proportional, and the highest capacitance was that of the TCP coating, the lowest that of the TCP + CED coating system, and the intermediate corresponds with the CED coating, as expected.

**Table 3.** Impedance parameters obtained for all examined samples in 0.5 M NaCl solution without (w/o) induced-damage incisions. The values are the averages from at least triplicate, as previously described.

| Parameters (without Incision) | TCP | CED | TCP + CED |
|---|---|---|---|
| Rs ($\Omega \cdot$cm$^2$) | 8.5 ($\pm$0.5) | 14 ($\pm$2) | 15 ($\pm$2) |
| $Z_{\text{CPE 1}}$ ($10^{-6}$ F/cm$^2$) | 2.8 ($\pm$0.06) | 0.96 ($\pm$0.03) | 0.12 ($\pm$0.06) |
| $R_1$ ($10^3$ $\Omega \cdot$cm$^2$) | 0.98 ($\pm$0.07) | 125 ($\pm$8.5) | 0.91 ($\pm$2.1) |
| $n_1$ | 0.80 | 0.57 | 0.91 |
| $Z_{\text{CPE 2}}$ ($10^{-6}$ F/cm$^2$) | 15.4 ($\pm$0.6) | 7.4 ($\pm$0.8) | 2.5 ($\pm$0.5) |
| $R_2$ ($10^3$ $\Omega \cdot$cm$^2$) | 15.5 ($\pm$0.5) | 187.5 ($\pm$22) | 543 ($\pm$18) |
| $n_2$ | 0.70 | 0.60 | 0.41 |
| W ($10^3$ $\Omega \cdot$cm$^2$) | 0.29 ($\pm$0.03) | 125 ($\pm$12) | 49 ($\pm$4) |
| $\chi^2$ | $198 \times 10^{-4}$ | $59 \times 10^{-4}$ | $86 \times 10^{-4}$ |
| Sum of Sqr. | 0.72 | 0.77 | 0.99 |

In Table 4, the impedance parameters of the samples with induced-damage incisions on the surface are shown. The same trend is observed when Table 3 and potentiodynamic polarization curves were analyzed. This demonstrates that incisions do not modify the corrosion resistance tendencies provided, i.e., TCP < CED < TCP + CED.

Comparison with investigations previously reported in the literature is a difficult task due to the several distinct parameters applied. However, when the EIS results of the TCP samples is compared with a previous study developed by Tian et al. [10], who also examined a steel bare coated tricationic phosphate, independent of the operational parameters, same order of magnitude for $Z_{\text{CPE 1}}$ and $Z_{\text{CPE 2}}$ compared with $\text{CPE}_c$ and $\text{CPE}_{dl}$ were attained. Although in the studies of Tian et al. [10] no Warburg component was utilized, similar significances of the EIS parameters permit this comparison. Moreover, interestingly, the results corresponded with a sample coated by a TCP method, and a similar corrosion current density obtained by Tian et al. [10] is also verified in this present investigation (i.e., ~9.86 $\mu$Acm$^{-2}$ in Tial et al. [10] study, while in this present investigation it was ~10.2 $\mu$Acm$^{-2}$).

An important comparison is one between the EIS results of the examined sample with and without provoked incisions. Thus, when comparing Table 3, which contains EIS parameters corresponding with the samples without incisions provoked, it was clearly observed that the resulting corrosion resistances of the samples with incisions are substantially decreased. For instance, in a general way, the capacitances ($Z_{\text{CPE 1}}$ and $Z_{\text{CPE 2}}$) were increased, while both $R_1$ and $R_2$ were decreased. When $R_1$ is analyzed, which corresponds with the resistance of the outer layer, the TCP and CED samples decreased more than 10$\times$, while the TCP + CED sample was not significantly modified. This is evidence that the thickness and nature of the involved coating systems had no substantial holes in their reactions at the coating/electrolyte interface when an incision is provoked. On the other hand, when $R_2$ is analyzed, the decrease is of about 15$\times$ when the TCP coating is considered, and both CED and TCP + CED coating systems have decrease in the same order of magnitude (~117$\times$ and 85$\times$, respectively).

**Table 4.** Impedance parameters obtained for all examined samples in 0.5 M NaCl solution considering induced-damage incisions. Values are averages from at least triplicate, as previously described.

| Parameters (with Incision) | TCP | CED | TCP + CED |
|---|---|---|---|
| Rs ($\Omega \cdot cm^2$) | 5 ($\pm 0.5$) | 5 ($\pm 2$) | 15 ($\pm 2$) |
| $Z_{CPE\,1}$ ($10^{-6}$ F/cm$^2$) | 246 ($\pm 0.3$) | 0.015 ($\pm 0.08$) | 19.1 ($\pm 1.2$) |
| $R_1$ ($10^3$ $\Omega \cdot cm^2$) | 0.085 ($\pm 0.02$) | 9.5 ($\pm 0.2$) | 1.1 ($\pm 0.15$) |
| $n_1$ | 0.68 | 0.98 | 0.90 |
| $Z_{CPE\,2}$ ($10^{-6}$ F/cm$^2$) | 96 ($\pm 36$) | 192 ($\pm 0.2$) | 36 ($\pm 1.7$) |
| $R_2$ ($10^3$ $\Omega \cdot cm^2$) | 0.98 ($\pm 0.02$) | 1.6 ($\pm 0.02$) | 6.4 ($\pm 0.1$) |
| $n_2$ | 0.85 | 0.69 | 0.70 |
| W ($10^3$ $\Omega \cdot cm^2$) | 0.035 ($\pm 0.002$) | 5.2 ($\pm 0.7$) | 20.6 ($\pm 5$) |
| $\chi^2$ | $48 \times 10^{-4}$ | $82 \times 10^{-4}$ | $51 \times 10^{-4}$ |
| Sum of Sqr. | 0.58 | 0.99 | 0.60 |

It is remembered that the resistance $R_2$ is associated with an inner layer of the coating interfacing CED with bare steel bare (designated as the CED sample) and CED with steel coated with a TCP layer (designated as TCP + CED sample). Since the TCP has a lower thickness (up to 1.5 μm), and its typical morphology is porous, this coating constitutes a substrate–coating interface, while the CED and TCP + CED samples are thicker (~15 and 16 μm) than TCP and is similar between them. Besides, the fact that the top coatings are equal, i.e., CED, the electrolyte penetration due to cross-link density was decreased and the micro-sized porous provoked [20,24] seems to provide this more intense reduction of $R_2$ values. It is remarked that the coating thickness is important to compare similar coatings. For example, it is not correct to compare the TCP and CED or TCP + CED. Firstly, the nature, morphology and thickness of TCP and CED are very distinctive. Evidently, this will favor coating with CED layers, which will demonstrate a better corrosion resistance. When same natures of coating are compared, the thickness has its importance level. Acamovic et al. [25] have reported that a CED coating on steel bare containing thickness of 18 μm (immersed into a 3% NaCl solution for 18 days) has a higher corrosion resistance than others containing only 10 μm.

When the role of the TCP layer in a coating system is analyzed, it is evidenced that it is only a TCP layer on steel bare, which is independent if an incision is provoked; the lowest corrosion resistance (evaluated by polarization and EIS techniques) is that of the TCP sample. As expected, this seems to be intimately associated with porous material and is thinner than other coatings with CED layers. On the other hand, when the CED and TCP + CED coating systems are compared, the attained results favor the sample with the substrate coated with a TCP layer.

Although similar thicknesses are reached for the CED and TCP + CED samples, the initial reactions and its corresponding evolutions indicate that the TCP + CED coating system sample has the porous electrode behavior well characterized in a certain frequency range. Observing the experimental Nyquist plots (Figure 5), at frequency ranges between $10^5$ Hz up to <$10^3$ Hz, the porous electrodes corresponding with the TCP + CED sample is evidenced (Figure 5b). The CED samples have porous electrode characteristics up to ~10 Hz. Between 10 Hz and 0.25 Hz, the straight line forming 22.5° suggests that the TCP + CED sample has a complex intermediate species acting as a barrier or as quasi-barrier protection.

When the samples with incisions are evaluated, the role of the TCP layer protection is fundamentally dismissed. This is due to the substrate being evidently exposed. It is expected that all samples examined present similar reactions. In fact, in all the three samples examined, considering a high frequency domain up to ~10 Hz, porous electrode behavior is characterized. At 10 and 0.25 Hz, the highest branch and semi arc segments are that of the TCP + CED sample, followed by the CED and TCP samples, respectively (Figure 6a,b).

### 3.4. SEM Micrographs, EDX and XRD Measurements

Figure 8a shows typical SEM (scanning electron microscope) images TCP + CED coating system samples with induced-damage incision. The top layer is CED coating on a TCP layer, which is covering the steel bare. This last layer constitutes the substrate (CRS, cold rolling steel sheet), as depicted in high magnification in Figure 8b. The CED layer is covering all surfaces of the examined sample. Evidently, its surface morphology is slightly rough as a typically applied based-epoxy coating, which is demonstrated in Figure 8c. The tricationic phosphate (TCP) is located between the CED layer and the substrate and is constituted by small Zn phosphate crystals sized between 3.5 and 6.0 μm, as shown in Figure 8d,e. No individual SEM images of the CED and TCP samples are depicted due to these being very similar to those shown in Figure 8.

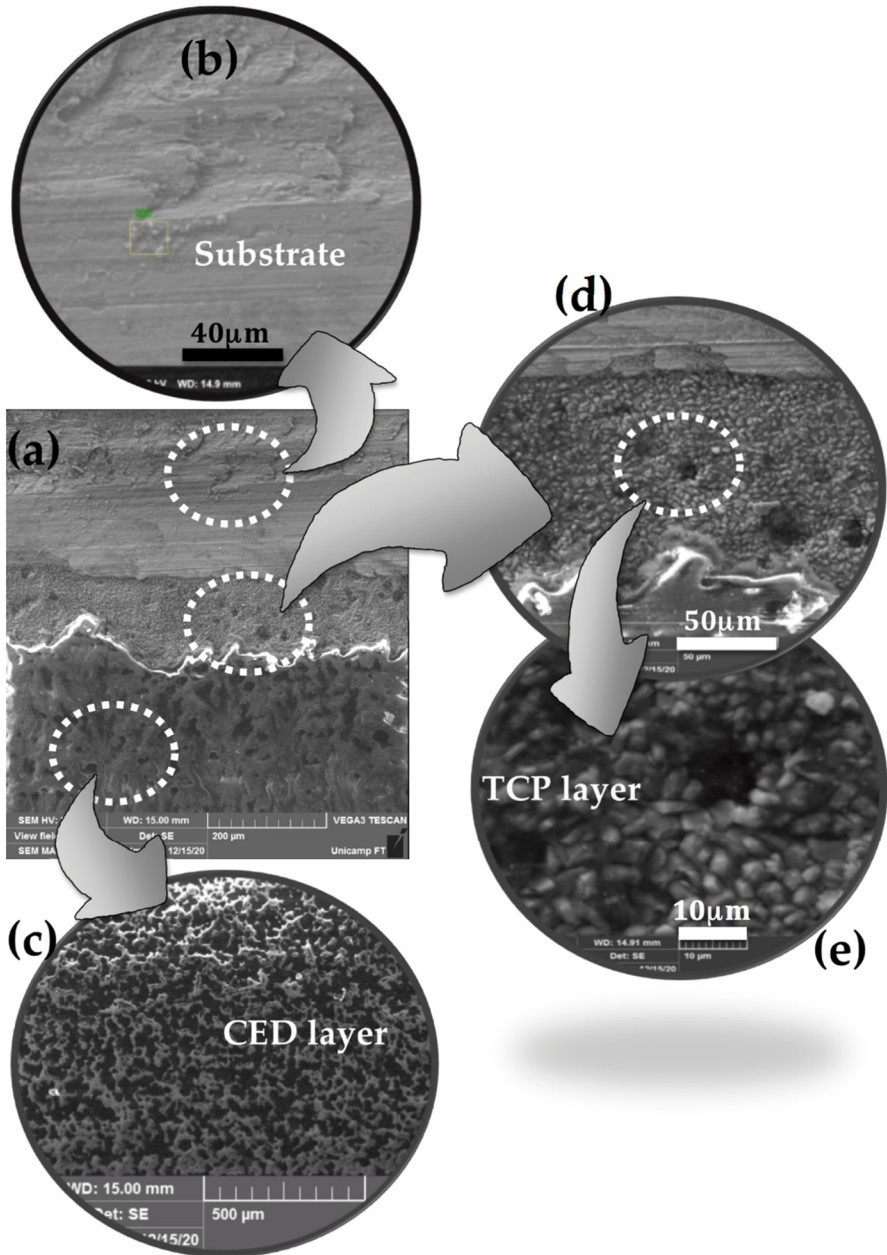

**Figure 8.** (**a**) Typical SEM micrographs of induced-damage incision coating system, evidencing (**b**) substrate (bottom), (**c**) the CED layer (on top), and (**d**,**e**) the TCP layer depicting Zn phosphate crystal particles.

The experimental results of the EDX (energy dispersive x-ray analysis corresponding with a typical analysis carried out in incision into the TCP + CED coating system sample) is shown in Figure 9a. Figure 9b depicts the typical peaks corresponding with the EDX analysis of the CRS (steel substrate). The main counts per second (cps) corresponding with iron are characterized at ~0.7, ~6.4 and ~7 keV, as expected.

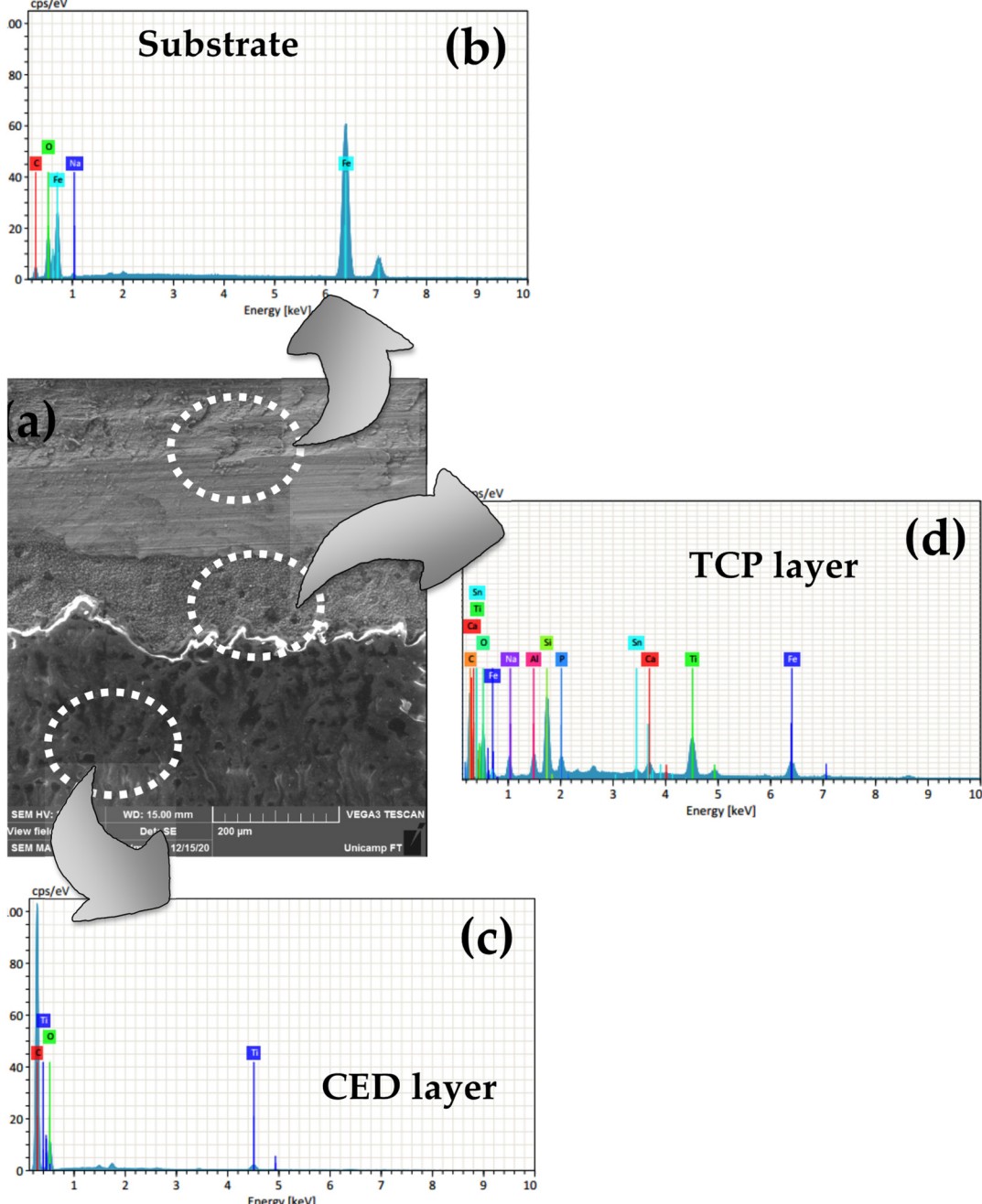

**Figure 9.** Experimental results of the EDX (energy dispersive x-ray analysis) of (**a**) the TCP + CED coating system sample showing an EDX analysis of: (**b**) steel substrate, (**c**) the CED layer and (**d**) the TCP layer.

Figure 9c depicts the main peaks corresponding with the CED layer. As was also expected, a based epoxy coating mainly peaks with associated carbon and Ti. This latter is identified due to the $TiO_2$ present in composition to provide white color in coating. Both Sn and Si act as catalyzer agents, and other elements are mainly constituents of the bathing

composition to constitute a TCP layer. In Figure 9d, the analysis is majority carried out at the TCP layer, and the main element constituents of the TCP layer and Fe ions are also identified, as aforementioned.

Although EDX analysis provides information concerning element contents in each layer of the coating system, the analyses of XRD (x-ray diffractogram) patterns are carried out. The three distinct coating system samples are evaluated, i.e., TCP, CED, and TCP + CED samples, as shown in Figures 10a, 10b and 10c, respectively. These analyses were carried out in examined samples that contain incisions exposing a substrate. These measurements were made before the EIS and polarization measurements. Before XRD measurements, the samples were cleaned using distilled water and were dried to remove the excessive portion of NaCl (electrolyte). Since all of the examined samples are damaged, all of the main Fe intensity peaks were detected. In this same trend, all samples exhibited peaks associated with FeOOH products, indistinctively, if different species of FeOOH designated as goethite, akaganeite and lepidocrocite were detected. Lepidocrocite (JCPDS 44−1415) and akaganeite (JCPDS 34−1266) were commonly detected [41–43].

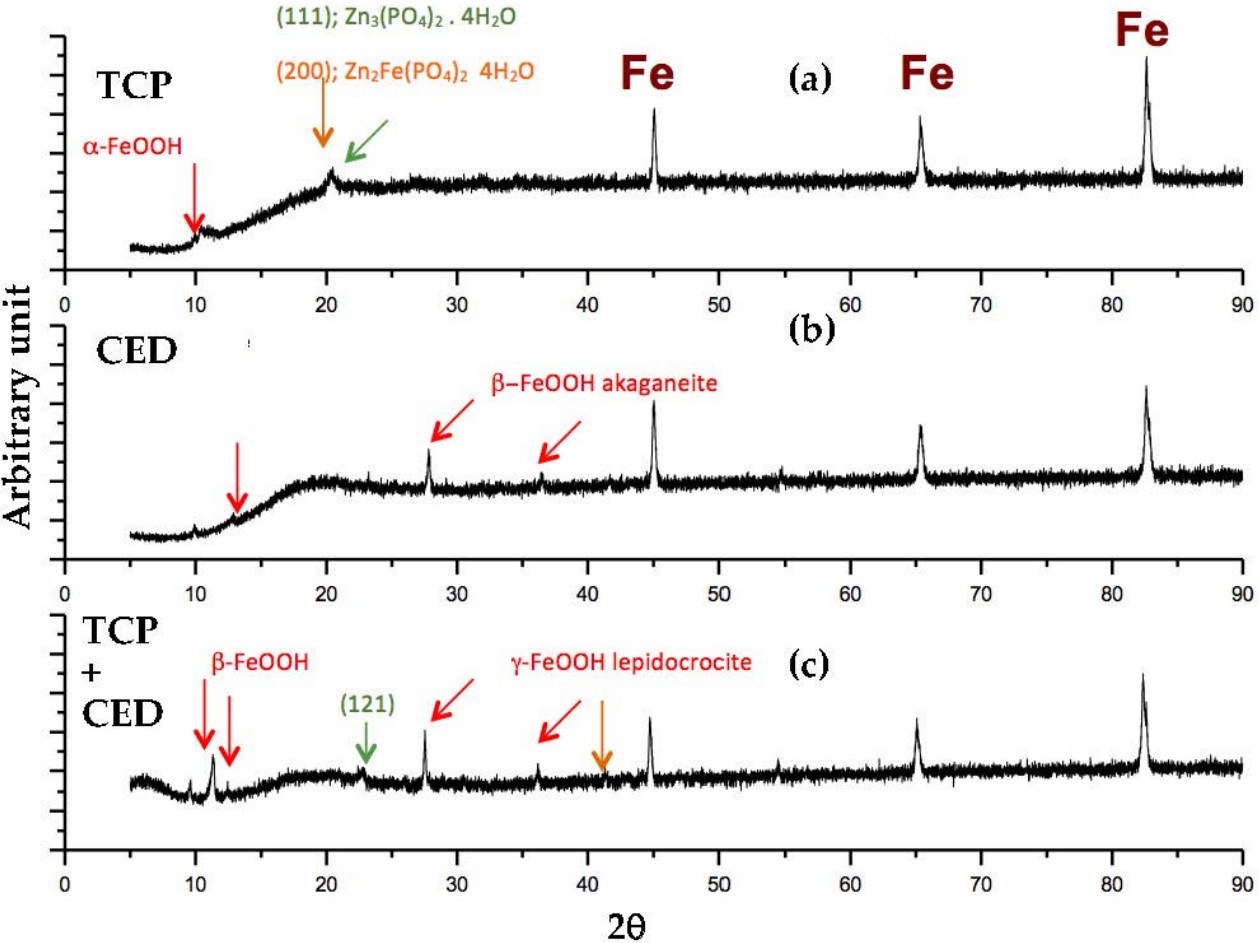

**Figure 10.** Typical XRD patterns of (**a**) the TCP coating system sample, (**b**) the CED sample and (**c**) the TCP + CED coating system. The JCPDS numbers are 37−0465 and 29−1427 for hopeite and phosphophyllite and 44−1415 and 34−1266 for lepidocrocite and akaganeite, respectively.

The $Zn_3(PO_4)_2 \cdot 4H_2O$ and $Zn_2Fe(PO_4)_2 \cdot 4H_2O$ phases are commonly designated as hopeite (JCPDS 37−0465) [10] and phosphophyllite (JCPDS 29−1427) [10] is identified when TCP and TCP + CED coating samples are analyzed. These identified phases are those intermediates absorbed by species at the interfaces analyzed. Evidently, these species have important roles on the evolution of the electrochemical behavior. As the evolution of the corrosion process, electrolytes penetrates throughout the hole-defects and porosity. Thus,

the substrate is reached and these species play as corrosion by-products, acting without effective protection; due to transport and diffusional phenomena, certain barrier protection can be achieved, as depicted in Figure 7.

### 3.5. EIS Measurements: Effect of Incision Conjugated with HDG Coating

Since it was discussed and verified that the effect of the TCP layer concatenated with incision in the resulting electrochemical behavior, the effect of hot dip galvanized (HDG) steel in order to constitute a multi-coating system (i.e., HDG/TCP + CED) is examined. This decision is based on the fact that in industrial applications, the HDG-coated steel substrate has been given great attention and utilization [13,18,19,44]. Analogue to those results previously discussed, firstly the potentiodynamic polarization, the number of time constants and Bode, Bode-phase and Nyquist plots were evaluated. As expected, by adding a new layer to constitute a new multi-coating system, the resulting thickness is increased. Consequently, due to thicker coating, it is physically predictable that the highest corrosion resistance is achieved. In order to verify this prediction, the coated HDG substrate is analyzed using only a TCP layer and TCP + CED layer, similarly to that made when the substrate without an HDG layer was evaluated.

Figure 11a shows the experimental result of the potentiodynamic polarization curves of the four distinct coating systems, i.e., the HDG, HDG/TCP, HDG/CED and HDG/TCP + CED samples. These were examined in a stagnant and naturally aerated 0.5 M NaCl solution at an environmental temperature. All samples also contain incisions to damage the coating systems. It is clearly observed that the lowest corrosion current density ($i_{corr}$) is that of the HDG/TCP + CED coating system. The highest $i_{corr}$ values are those of the CRS steel bare (no coating), i.e., 65 ($\pm$3) $\mu Acm^{-2}$, together with the HDG (steel bare) sample, i.e., 60 ($\pm$3) $\mu Acm^{-2}$. Based on these results, it is clarified that using only the HDG coating system, no effective protection is provided. Tsai et al. [44] have reported similar values of $i_{corr}$ and $E_{corr}$ when a sample constituted by steel-coated HDG is examined (i.e., ~55 $\mu Acm^{-2}$ and $-1129$ mV, respectively).

When the TCP layer is used on the substrate coated with HDG, the $i_{corr}$ decreases by about 10$\times$, attaining about 5 $\mu Acm^{-2}$, as depicted in Figure 9a. When the CED layer is applied onto the HDG sample, the $i_{corr}$ decreases by 2$\times$ when compared with that of HDG/TCP layers. Interestingly, when a multi-coting system is used, i.e., a steel substrate coated with HDG, followed by TCP and CED layers, the resulting $i_{corr}$ reaches approximately 1.1 ($\pm$0.3) $\mu Acm^{-2}$. This value is about 60$\times$ lower than uncoated steel substrate when compared with HDG-coated samples.

Considering the best result of the sample with the substrate first coated with an HDG layer, i.e., HDG/TCP +CED, and comparing it with the best result of that the sample containing only a TCP + CED layer (shown in Figure 2b), it is clearly observed that the HDG coating has a slight advantage over the TCP + CED sample. The HDG/ TCP + CED coating system has an $i_{corr}$ ~2 times lower (~1 $\mu Acm^{-2}$) than the TCP + CED sample (~2 $\mu Acm^{-2}$). However, it is verified that the HDG/TCP + CED sample has a corrosion potential considerably displaced to the more active potential side, i.e., $-1129$ mV against $-435$ mV (vs. SCE). This has a low deleterious effect when compared with the attained corrosion current density, which seems to be more drastic in terms of the severity of the degradation. Another interesting part of the attained results concerns the primary passive current ($i_{pp}$), which is demonstrated by the coating system with the HDG layer. This $i_{pp}$ firstly occurs for the HDG/TCP + CED followed by the HDG/CED sample. Both the HDG sample and the HDG/TCP sample reveal an $i_{pp}$ about 10$\times$ higher than other two samples. This result favors both the HDG/TCP + CED and HDG/CED samples. This passivation seems to be correlated with the diffusional and transport of intermediate species, which seem to provide some barrier protection when these are allocated inside the porous structure in the TCP layer and hole-defects at the CED layer.

When Figure 9b,c is analyzed, it is revealed that two time constants are prevalent and the modulus of impedance and the maximum phase angle corresponding with the

multi-coating system HDG/TCP + CED samples exhibits the best result. When analyzing Nyquist plots in Figure 9d,e it is confirmed that the corrosion resistance tendency is HDG < HDG/TCP < HDG/CED < HDG/TCP + CED. This is similar to that trend observed when steel substrates without an HDG coating were examined. This indicates that only by applying a TCP layer, or only a CED layer, on HDG-coated steel is not enough to provide the best corrosion resistance.

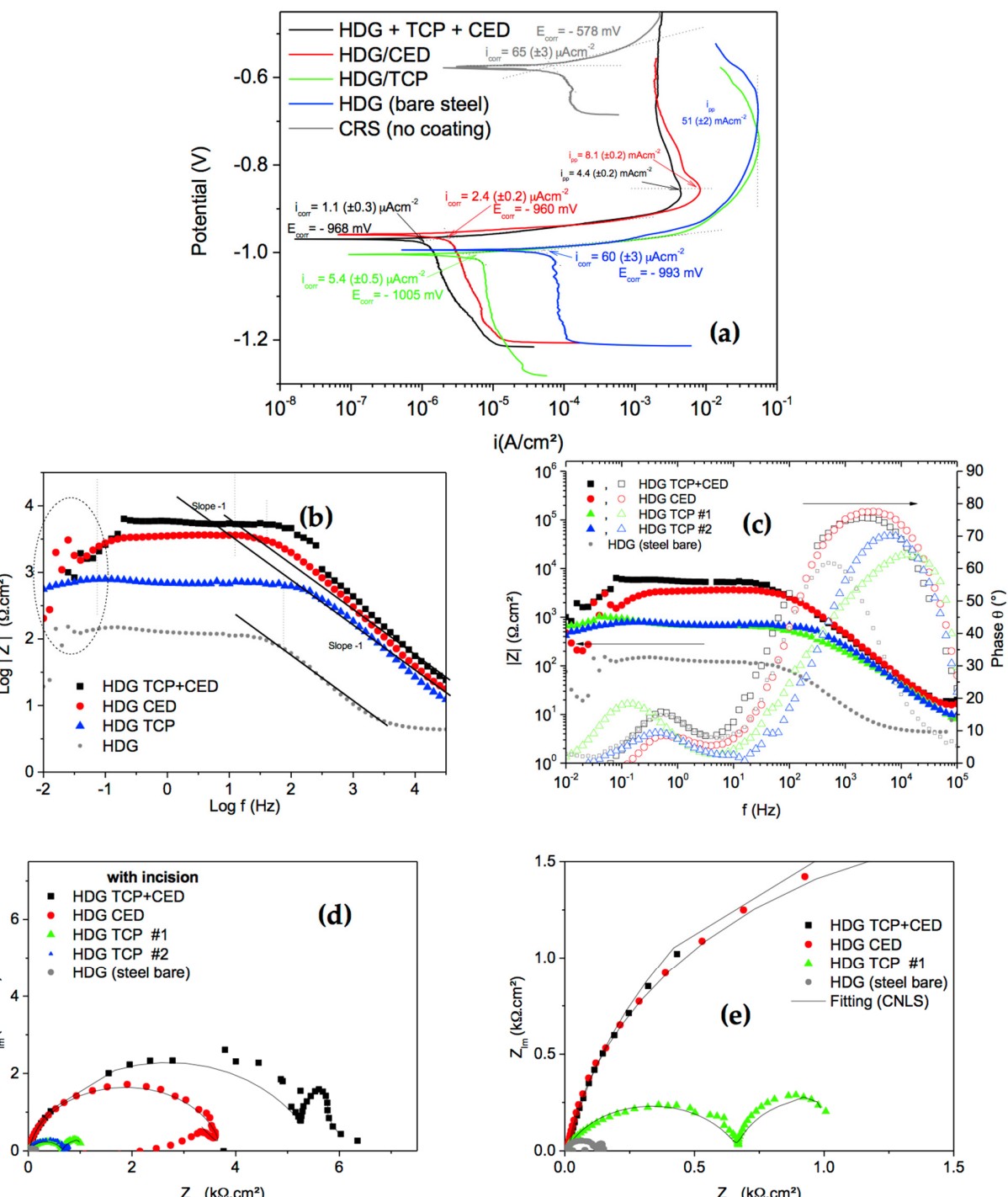

**Figure 11.** Results of (**a**) potentiodynamic polarization curves; (**b**) moduli of the imaginary parts of the impedances with frequencies; (**c**) Bode and Bode-phase plots; (**d**) Nyquist representations; and (**e**) details of Nyquist plots in a minor range of analysis of the four distinctive coating systems, i.e., the HDG, HDG/TCP, HDG/CED and HDG/TCP + CED samples, examined in 0.5 M NaCl solution at an environmental temperature. All samples contain incisions to damage the coating systems.

Another interesting obtained result concerns the porous electrode behavior. Both the HDG/CED and HDG/TCP + CED sample, which have demonstrated the best results of corrosion resistances, have no porous electrode behavior characterized. It is remembered that this behavior is revealed when at a high frequency domain range, a straight line forming 45° is constituted. This characteristic is only verified when the HDG/TCP sample and only steel coated with HDG are analyzed. This ensures that full protection is provided. This seems to be intimately associated with porous morphologies as intrinsic characteristics of these layers. With this, the water and electrolyte penetrate throughout layers and the substrate is reached and drastically attacked.

When the Nyquist plots corresponding with the HDG/CED and HDG/TCP + CED samples are analyzed, at a high frequency domain range, a quasi-ideal capacitive behavior (Bode-phase tending 90°) is verified, as was also observed by Reichinger et al. [19], Vakili et al. [20], Tian et al. [39], and Zhang et al. [45]. Comparing the HDG/TCP + CED coating system with the TCP + CED coating sample, two important assertions can be made. Firstly, it is the fact that the TCP + CED sample has a thickness (~7 $\pm$ 2 μm) that is considerably lower than the HDG/TCP + CED sample (~12 $\pm$ 3 μm). When it is verified that there are similar thicknesses between the TCP + CED and the CED samples, their corresponding resistances were very distinctive, which was attributed to the nature of the layers used and the resulting intermediate species formed, potentially protecting substrate degradation or retarding the corrosive process.

Based on these assertions and comparing the HDG/TCP + CED with the TCP + CED coating sample, i.e., Figure 6a with Figure 11d, it is qualitatively verified that these samples have similar sizes of the depressed semi arcs constituted ($Z_{Im}$ ~2 kΩ·cm$^2$ with $Z_{Re}$ ~6 kΩ·cm$^2$). A noticeable difference is that the sample without the HDG layer on a steel substrate reveals a porous electrode behavior corroborating with planar electrode behavior to prescribe the resulting electrochemical behavior. This indicates that an additional HDG layer significantly increases the thickness (~2 times) of the coating system. Considering the same short-term immersion period (which were evaluated in the samples), the sample with a thicker coating system provides a higher time to electrolyte penetration and to the substrate to be reached than the other one. Furthermore, the nature and electrochemical contribution of the distinctive intermediate species and corrosion by-product formed should also be taken into account. For example, Amirudin and Thierry [42] have reported that a sample with an HDG layer has Zn ions, locally modifying the pH and interacting with chloride ions, provoking a certain delamination. More recently, it has been reported [18] that a pitting corrosion on HDG-coated steel in short exposures has important consequences for the assessment of the lifetime of the zinc coatings on steel. Reichinger et al. [19] have reported that ion transport in HDG steel is conspicuously different from a cataphoretic coating.

With these observations, it is very important to analyze the quantitative results provided by EIS parameters. In this sense, the same equivalent circuit (EC) was utilized to obtain the EIS parameters of the samples without an HDG layer also being used. Figure 12 shows the schematic representation of the EC corresponding with each region of the coating system established. An equivalent circuit with a Warburg component is also depicted and intermediate species corroborate to predict the electrochemical corrosion behavior. When an electrolyte reaches the HDG layer, as previously reported [19,41], delamination can occur and intermediate species containing Zn corrosion by-products will also participate in the corrosion mechanism, and a certain "protection" can be provided. This can reasonably be understood when EIS parameters obtained by using CNLS simulations are analyzed, as shown in Table 5. When comparing EIS parameters among the coated HDG samples, it is evidenced that the same corrosion resistance tendency verified for those samples without an HDG layer (also containing incisions) is also verified, i.e., HDG < HDG/TCP < HDG/CED < HDG/TCP + CED.

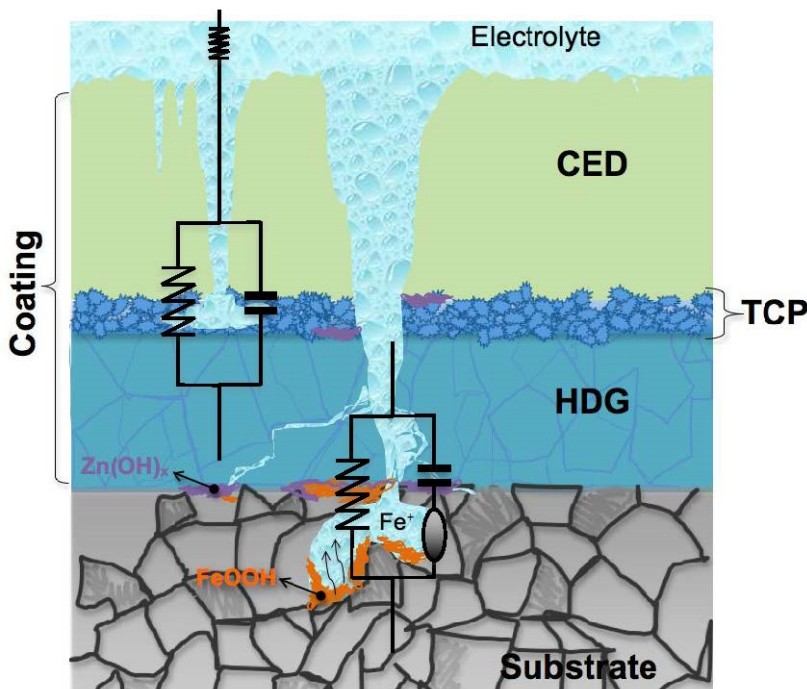

**Figure 12.** Schematic representation of a multi-coating system with the substrate being firstly coated with an HDG layer followed by other layers.

**Table 5.** Impedance parameters obtained for the samples containing an HDG layer when immersed into a 0.5 M NaCl solution.

| Parameters (with Incision) | HDG/TCP | HDG/CED | HDG/TCP + CED |
|---|---|---|---|
| Rs ($\Omega \cdot cm^2$) | 4 ($\pm 1$) | 11 ($\pm 0.5$) | 14 ($\pm 2$) |
| $Z_{CPE\ 1}$ ($10^{-6}$ F/cm$^2$) | 5.3 ($\pm 0.3$) | 0.64 ($\pm 0.04$) | 0.89 ($\pm 0.03$) |
| $R_1$ ($10^3$ $\Omega \cdot cm^2$) | 0.67 ($\pm 0.07$) | 3.4 ($\pm 0.03$) | 5.3 ($\pm 0.07$) |
| $n_1$ | 0.77 | 0.93 | 0.90 |
| $Z_{CPE\ 2}$ ($10^{-6}$ F/cm$^2$) | 2357 ($\pm 258$) | 1.67 ($\pm 0.5$) | 576 ($\pm 55$) |
| $R_2$ ($10^3$ $\Omega \cdot cm^2$) | 0.42 ($\pm 0.07$) | 0.33 ($\pm 0.02$) | 12.3 ($\pm 0.6$) |
| $n_2$ | 0.98 | 0.99 | 0.54 |
| W ($10^3$ $\Omega \cdot cm^2$) | 1.14 ($\pm 0.2$) | 38.5 ($\pm 5$) | 69.8 ($\pm 8.8$) |
| $\chi^2$ | $93 \times 10^{-4}$ | $44 \times 10^{-4}$ | $141 \times 10^{-4}$ |
| Sum of Sqr. | 0.99 | 0.45 | 1.21 |

Analyzing all parameters in Table 5, it can be understood that there is a corrosion mechanism proposed and schematically represented in Figure 12, that is also associated with those previously reported in the literature [41]. For instance, when the results of the HDG/TCP sample are examined, firstly, it can be seen that capacitance $Z_{CPE2}$ is higher than $Z_{CPE1}$ and $R_2$ (at HDG/TCP interface) is slightly lower than $R_1$ (at substrate/HDG interface). It is recognized that as capacitance increases, its correlated thickness is also decreased. This physically means that hole-defects are being prevalently permitting the electrolyte penetration and reaching the substrate. This provides a low resistance—$R_1$—while $R_w$ has a discreet effect upon the electrochemical behavior (the lowest value, i.e., 0.035 k$\Omega \cdot cm^2$). The sum of the three resistances (i.e., $R_1 + R_2 + R_w$) attains the lowest result when compared between the HDG/CED and the HDG/TCP + CED samples. On the other hand, the highest results of $R_1 + R_2 + R_w$ (~27 k$\Omega \cdot cm^2$) is that of the HDG/TCP + CED sample. In this case, the resistance $R_w$ has an important role in corrosion resistance. This is associated with the fact that distinct intermediate products and species corroborate in order to minimize the drastic corrosive effect. At least, the electrolyte penetration onto the steel substrate is retarded.

When compared with other HDG-coated samples, both $R_2$ and $R_w$, which are representing reactions at the substrate–HDG interface and at complex corrosion by-products inside degraded regions (pites), are higher than other examined samples.

Additionally, it is in doubt that the HDG layers affect the resulting corrosion resistance. In this sense, Tables 4 and 5 are compared. When the TCP sample is compared with the HDG/TCP, the $R_1$ (TCP) is lower than the HDG/TCP, while $Z_{CPE1}$ is increased. Using Figure 12 without a CED layer and comparing it with Figure 7a means that, at a similar immersion period, the TCP sample has poor protection or a higher penetration than the HDG/TCP sample. Moreover, when $R_2$ (TCP) is analyzed, which means reactions at the substrate–TCP and substrate–HDG interfaces, respectively, the TCP sample has higher values (~0.98 k$\Omega$·cm$^2$) than HDG/TCP (~0.42 k$\Omega$·cm$^2$). This means that the porous structure induces a decrease in the resistance at this interface. Furthermore, due to the decrease verified at $R_2$, and the intermediate species formed, the $R_w$ of the HDG/TCP sample (~1 k$\Omega$·cm$^2$) is higher than the TCP sample (~0.035 k$\Omega$·cm$^2$). This represents an interaction between intermediate species, probably those resulting in the complex products, generically represented by $ZnX_a(OH)_y$, as reported [18,19,42–44]. It is clearly observed that $R_1$ and $R_2$ and $Z_{CPE1}$ and $Z_{CPE2}$ vary according to corrosion progress. The sum of the three resistances can also be used to summarize the corrosion resistance trend. Thus, the HDG/TCP samples have an $R_1 + R_2 + R_w$ of about 2 k$\Omega$·cm$^2$, against ~1 k$\Omega$·cm$^2$ observed for the TCP sample.

Analogue evaluations can also be carried out for all of the examined samples. Considering that the sum of $R_1 + R_2 + R_w$ is verified and that those samples are firstly HDG-coated show that the results demonstrate a better corrosion resistance than the sample without an HDG layer application, i.e., the HDG/CED attains ~40 k$\Omega$·cm$^2$ and the CED ~17 k$\Omega$·cm$^2$; and the HDG/TCP + CED reaches ~85 k$\Omega$·cm$^2$, while the TCP + CED sample reaches ~30 k$\Omega$·cm$^2$. With these results, the previous question concerning the similar Nyquist plots between the HDG/TCP + CED and the TCP + CED coating sample is answered.

However, another question also remains concerning the light weight and relative cost associated with effective protection. Moreover, an additional question remains concerning the validations of these impedance parameters and potentiodynamic polarization values with traditional and conventionally salt spray results commonly utilized in industrial practices. Based on these questions, the next section is proposed.

### 3.6. Comparisons: Electrochemical, Salt Spray, Relative Cost Results and Effective Protection

Although the experimental results are indicative that a same trend in terms of the corrosion resistances of the examined samples are prevalent, indistinctively, if the HDG layers and incisions are considered, it is remarked that the short-term immersion period seems to constitute a limitation in this present investigation. This is adopted in order to verify and to compare all of the proposed samples. In the literature, there exist articles showing that same corrosion resistance tendency is kept after distinct immersion periods are adopted [15,20,39,45,46]. It is commonly verified that kinetics are alternated. For instance, in studies developed by Vakili et al. [20] using a modified Zn phosphate sample, and Noodeh et al. [15] using silica as nano particles modifying the cathodic electrocoating, it is observed that the moduli of impedances in a short-term immersion period is lower than the longer period, as expected. However, similar trends are verified. Interestingly, what is also modified is the porous electrode behavior in the electrochemical process. For example, in the sample that was immersed over 7 days, only planar electrode behavior domains consisted of the electrochemical process, and after 14 days, porous electrode behaviors corroborate to predict the corrosion response. Evidently, the distinct nature of coating also modifies the participation of porous electrode behavior in corrosive processing [10]. The chemical nature of the electrolyte also drastically modifies the porous electrode behavior in an electrochemical process [44].

In a general way, and also based on the experimentations observed, it can be said that, at initial stages, the intermediate species should contribute with minimum effects. Firstly,

this is due to the substrate not being attacked, and because no completed penetration is reached. With the immersion period increasing, these species have their roles initiating importance in corrosion evolution. With the electrolyte penetration and the substrate being reached, it is possible that a "temporally" oxide film barrier of protection is formed. Based on this, the aim in this present investigation is to evaluate the electrochemical behavior of the distinct multi-coating systems in the initial immersion period, i.e., up to 2 h of immersion. Although no other immersion periods were examined, it is believed that the corrosion tendency verified in a short-term immersion is also prevalent in long-term immersion periods, as verified in other distinct studies [15,20,45,46].

Although a comparison between EIS parameters, corrosion current densities and salt spray results is not practicable in terms of a direct quantitative parameter comparison, some investigations have reported both EIS and salt spray results, as reported by Shreepathi et al. [46], Noodeh et al. [15] and Tsai et al. [44]. It can be convenient to decide or to select a certain application or to provide an analysis of optimization costs. Based on this, all samples examined were also subjected to salt spray conditions. Tsai et al. [44] reported that salt spray results are in good consistence with both polarization and EIS results. In our results, it is confirmed that the same corrosion resistance tendency was verified using corrosion current density and EIS parameters, which is also observed when the salt spray results are analyzed.

Figure 13a depicts a typical sample with scribe marks and no immersion, which is representative of the other samples to be examined during 500 h (~21 days), following description in ASTM B 117 [44,46]. Figure 13b–d show the three distinct coating systems with different layers being TCP + CED on the substrate (CRS), and only a CED layer and only a TCP layer deposited on the CRS substrate, respectively. Additionally, the coating system samples with their substrate priory coated with HDG layers are shown in Figure 13e–g, which represent the sample HDG/TCP + CED, the HDG/CED and the HDG/TCP, respectively.

It is clarified that samples without an HDG layer clearly exhibit red rust formation, while only the HDG/TCP sample has no depicted white rust formation. Based on the images of a photograph corresponding with each examined coating system, it is clarified that the same corrosion resistance tendency is verified, i.e., TCP + CED > CED > TCP, and when HDG layers on CRS substrates are considered, the verified trend is HDG/TCP + CED > HDG/CED > HDG/TCP. When the corrosion current density and the EIS parameters were analyzed, the same trend was also observed. These observations indicate that EIS and polarization techniques are helpful and powerful tools to predict the corrosion resistances of distinct coatings, which makes it possible to replace the accelerated salt spray measurements.

Although the salt spray results during the 500 h indicate the same tendency of corrosion resistance to that verified by electrochemical measurements, there remains a question of concerning operational/manufacturing costs, coating weights and effective protection provided for each distinct coating system. The protection efficiency ($\eta$) can be determined by using Equations (2) and (3) [39,45], when values of $i_{corr}$ and the sum of resistances (EIS parameters) are used, respectively:

$$\eta_{(icorr)} = (i_{corr(Bare)} - i_{corr})/i_{corr(Bare)}) \times 100\% \tag{2}$$

$$\eta_{(R)} = (\Sigma R_{(Bare)}) - \Sigma R)/\Sigma R_{(Bare)}) \times 100\% \tag{3}$$

where $i_{corr(Bare)}$ and $i_{corr}$ are the experimental corrosion current densities of the CRS (steel substrate) and coating systems examined, respectively. At Equation (3), $\Sigma R_{(Bare)}$ and $\Sigma R$ represent the sum of resistance $R_1 + R_2 + R_w$ of the substrate without coating and those examined coating systems, respectively. The results of the $\eta$ correlating $i_{corr}$ and $\Sigma R$ are $\eta_{(icorr)}$ and $\eta_{(R)}$ and are shown in Table 6. The values of $i_{corr}$ and $R_1 + R_2 + R_w$ are obtained from Figures 2 and 11, and Tables 4 and 5. The ranges of cost per area deposited ($\$/cm^2$) are estimated based on the commercial price used to obtain a covered sample of 300 ($\pm$5) $cm^2$.

Evidently, this $/cm$^2$ can be modified from a distinct country and its economy, which is induced to determine a relative cost (RC) based on the lowest attained range cost, i.e., the TCP sample. Since the relative cost is considered, it was also determined as a relative protection efficiency based on those $\eta_{(icorr)}$ considering the TCP sample. The $\eta$ results clearly favor the HDG/TCP + CED sample, followed by the TCP + CED, indistinctively, if $\eta_{(icorr)}$ or $\eta_{(R)}$ are considered.

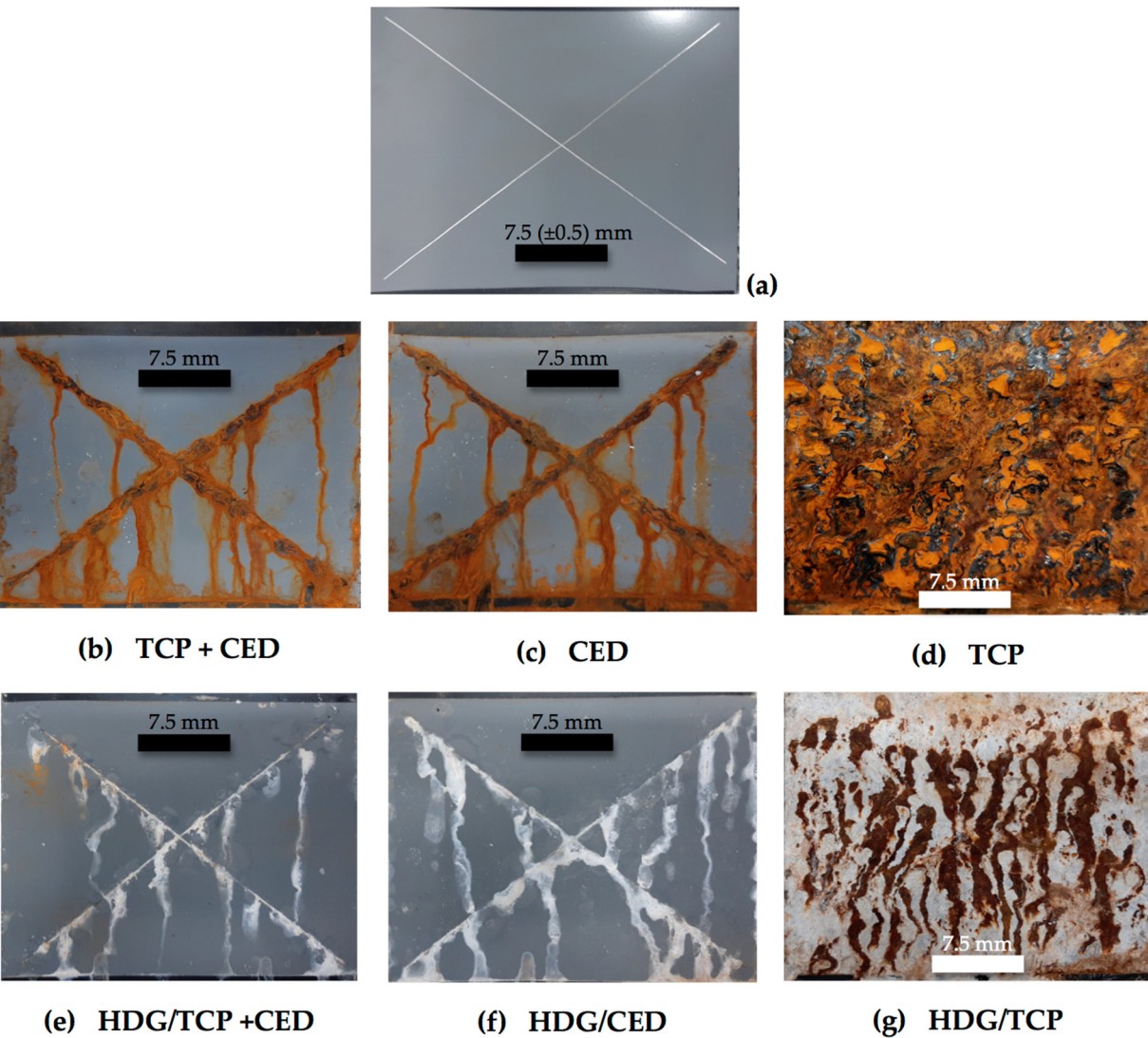

**Figure 13.** Typical resulting photographs of the coating systems examined: (**a**) no immersed sample showing scribe marks as ASTM B117; (**b**) the TCP + CED sample; (**c**) CED; (**d**) TCP sample coating; (**e**) HDG/TCP + CED; (**f**) HDG/CED; and (**g**) HDG/TCP samples exposed in NaCl solution during 500 h (~21 days), according to ASTM B 117. Samples without an HDG layer clearly exhibit red rust, while only the HDG/TCP sample has no depicted white rust formation.

However, there also remains another question concerning the thickness and weight provided by each one of the coating systems examined. The electrochemical results provide information of corrosion resistance for each kind of substrate, i.e., only CRS without covering and a CRS with an HDG layer. From Table 6, the attained results provide a comparison between the two different substrates. However, as has also been pointed out when analyzing EIS parameters, a sample with an increased thickness has a higher resistance than other ones with a lower thickness.

In order to compare and to understand the effect of the average values of coating weight (in g/m$^2$) and the coating thickness (in μm) for each different examined sample,

the ratio between the coating weight (CW) and coating thickness (CT) of these values are plotted as depicted in Figures 14a, 14b and 14c, respectively. It is clearly observed that the two distinct kinds of substrates (CRS and HDG) have different ranges of the CW. This is attributed to the fact that the highest CW is that of the HDG layer (~40 g/m$^2$). Additionally, S-shape type curves characterize the variations between two substrate families. However, different to the CW that exhibited a complementary trend, CT shows intermediate values of the CED and TCP/CED samples, as shown in Figure 14a,b.

**Table 6.** Parameters utilized to determine the protection efficiency (η) with respect to corrosion current densities (i$_{corr}$) and the sum of polarization resistances (R$_1$ + R$_2$ + R$_w$) of the examined samples. The range values of cost of each coating system are considered ($/cm$^2$) and these are relativized (RC, relative cost) with respect to the TCP sample. Bold and underlined values are the best results in each column.

| Sample | $/cm$^2$ | RC$_{(TCP)}$ | i$_{corr}$ (μA/cm$^2$) | R$_1$ + R$_2$ + R$_w$ (10$^3$ × Ω·cm$^2$) | η$_{(icorr)}$ (%) | η$_{(R)}$ (%) | η$_{(TCP)}$ |
|---|---|---|---|---|---|---|---|
| Steel Bare | 0.0035~0.0042 | 0.6 | 68 | 0.133 | - | - | - |
| HDG | 0.0109~0.0115 | 1.6 | 60 | n/a | 11.8 | n/a | 0.14 |
| TCP | 0.0020~0.0026 | 1 | 10.2 | 1.1 | 85 | 88 | 1 |
| CED | 0.0122~0.0156 | 2.2 | 3.71 | 17 | 94 | 99.2 | 1.11 |
| TCP + CED | 0.0142~0.0182 | 3.2 | **1.98** | 28 | **97** | 99.5 | **1.14** |
| HDG/TCP | 0.0129~0.0141 | 2.6 | 5.4 | 2.2 | 92 | 94.2 | 1.08 |
| HDG/CED | 0.0231~0.0271 | 3.9 | 2.4 | 42 | 96 | 99.7 | 1.13 |
| HDG/TCP + CED | 0.0251~0.0297 | 4.9 | **1.1** | **87** | **98** | 99.8 | **1.15** |

It is remarked that when a CRS sample (substrate) is covered with a TCP layer, its CW is ~2.5 g/m$^2$, and a CRS covered with an HDG layer has a CW of about 4 g/m$^2$ and the resulting CT is approximately 1 μm. Based on the aforementioned observations, there is still doubt concerning the cost, protection efficiency, and weight and thickness concatenated. With this, a relative comparison among values of the examined samples considering relative costs with respect to the TCP sample (RC$_{(TCP)}$) and its protection efficiency was also relative to the TCP sample (η$_{(TCP)}$), as shown in Table 7. This RC$_{(TCP)}$/η$_{(TCP)}$ parameter was also designated, as X reveals that the highest cost per efficiency is that of the HDG/TCP + CED sample, which has a higher operational cost and quantities of deposited layers, as expected.

Figure 14c depicts the results of efficiency protection (η) with coating density (d). This is calculated considering that (CW/CT) represents the coating density (in g/cm$^3$). When η is plotted as a function of d, three distinct ranges are characterized. First it is characterized with an efficiency higher than 95% and a density lower than 4 g/cm$^3$. With this, only three coating systems are potentially selected, i.e., the HDG/TCP + CED, HDG/CED and the TCP + CED samples. This indicates that no HDG layer ever provides a better efficiency protection with a lower density. From those examined samples, two coating systems demonstrate an efficiency between 95% and 90%, while the other two have an efficiency lower than 90%.

The X/d $_{(TCP)}$ parameter is also analyzed, which refers to an analysis concerning the relative cost per efficiency protection per relative coating density, as is also shown in Table 7. When X/d $_{(TCP)}$ values are analyzed, it is revealed that the two lowest values are those of the HDG/TCP + CED and HDG/CED coating system samples. However, considering possible error ranges in all involved parameters, it is interestingly verified that the TCP + CED sample has the same order of magnitude (i.e., 0.11 and 0.13).

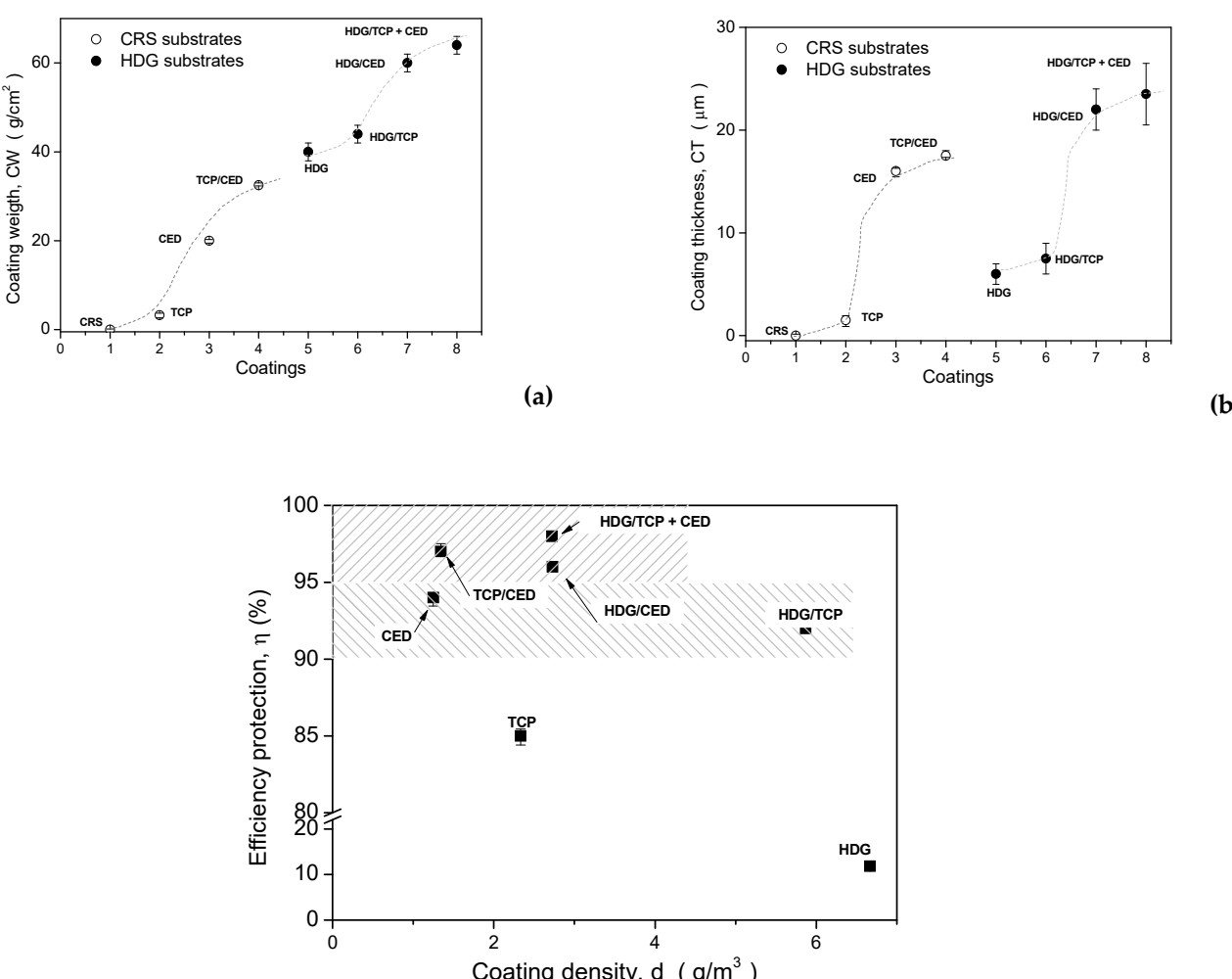

**Figure 14.** (**a**) Variation of coating weight (CW) and (**b**) the coating thickness (CT) and CW-to-CT ratio of the all examined coating systems involving two distinct substrate, i.e., CRS and HDG. (**c**) correlation of η with the coating density of the examined samples.

**Table 7.** Parameters utilized to determine the relative cost (RC) per relative protection efficiency (η) and per relative coating density with respect to the TCP sample. CW and CT are used to determined d (i.e., CW/CT). Bold and underlined values are the best results in each column.

| Sample | $RC_{(TCP)}/\eta_{(TCP)}$ X * | CT (μm) | CW (g/m²) | d (g/cm³) | $d_{(TCP)}$ () | $X/d_{(TCP)}$ |
|---|---|---|---|---|---|---|
| HDG | 11.4 | 6.0 | 40 | 6.67 | 2.86 | 3.98 |
| TCP | 1 | 1.5 | **3.5** | 2.33 | 1 | 0.43 |
| CED | 1.99 | 16.0 | 20 | 1.25 | 0.54 | 0.19 |
| TCP + CED | 2.80 | 17.9 | 24 | 1.34 | 0.58 | **0.13** |
| HDG/TCP | 2.40 | 7.5 | 44 | 5.87 | 2.51 | 0.16 |
| HDG/CED | 3.45 | 22.0 | 60 | 2.73 | 1.17 | **0.11** |
| HDG/TCP + CED | **4.25** | **23.5** | **64** | 2.72 | 1.17 | **0.09** |

* Parameter calculated by using relative cost based on the TCP sample per efficient protection (η), which is also relativized on the TCP sample values, designated as X parameter.

Based on the attained results, the roles of Zn phosphate and HDG on electrochemical behaviors of the coating systems containing these layers are confirmed. It was found that by only using the TCP layer and the HDG layer, the best protection is not reached. A similar conclusion is provided when only the CED layer is used. However, when the

TCP layer is deposited on an HDG layer followed by a CED layer, from the investigated coating systems, the highest efficiency is demonstrated (i.e., HDG/TCP + CED). This efficiency is followed by that sample without HDG coating but containing TCP + CED, which also demonstrates the great importance of the CED layer. However, this unique layer or concatenated HDG layer has not provided good results. It was also found that the porous electrode behavior also has a remarked importance to predict electrochemical responses and to help an adequate selection of the coating systems. A porous electrode behavior does not mean a deleterious aspect as it is commonly associated with planar electrode behavior. It seems that the thicker deposited layer is a crucial factor to constitute only a planar electrode behavior, but it is not thick enough. The nature and morphology of the deposited layer also has an essential importance. For instance, the HDG/CED coating has higher thicknesses than the TCP + CED coating system (i.e., 22 μm against 17.9 μm), but the HDG/CED has efficiency protection in the same order of magnitude (96 and 97%, respectively). Moreover, it is concluded that only samples containing HDG associated with an CED layer provide only planar electrode behavior, which suggests that a thickness greater than 22 μm provides predominant planar behavior to predict the electrochemical behavior of the examined coating systems. Similar results have also been attained when Ramezanzadeh et al. [24] have investigated coating systems with thicknesses reaching more than 60 μm (involving phosphate, CED and primer layers). Jegdic et al. [47] have also reported only planar electrode behavior when coating systems higher than 50 μm are investigated.

This is independent of the immersion period examined. Only after more than 128 days of an immersed period (into NaCl solution) does the sample slightly indicate both porous and planar electrode behaviors concatenated, i.e., a straight line at 45° in the Nyquist plot in a high frequency domain is characterized [47]. This is surely associated with the fact that polymeric coating has been considerably deteriorated. This is due to the decreased cross-link density and holes provoked on the surface that permit the substrate to be severely attacked [24]. In these two aforementioned studies [24,47], the primer layer is more intimately associated with mechanical properties than the corrosion behavior, and this with the CED layer deposited on the TCP layer. Furthermore, a better adhesion of the CED layer is attained when a phosphate layer is applied [24,47].

## 4. Conclusions

From the attained experimental results, the following conclusions can be drawn:

- The roles of Zn phosphate and HDG layers on electrochemical responses of the coating systems are elucidated. It is found that only Zn phosphate and only HDG layer depositions have not provided the best corrosion protection. When a TCP layer and HDG layers are applied and followed by a CED layer, the highest efficiency is attained.
- The conjugated porous and planar electrode behaviors do not mean that a deleterious effect on protection efficiency occurs. The existence of porous electrode behavior is more intimately associated with the initial thickness coating, while corrosion resistance is associated with adhesion of the CED layer on the TCP coating, which is provided by this latter layer.
- When the relative cost-to-efficiency relative to the coating density ratio is evaluated, the best results do not correspond with those thicker deposited samples, or with those that do not contain a porous electrode behavior, but with those that contain at least two layers constituting a thickness greater than 20 μm. Additionally, if a CED layer is obligatory for a subsequent primer and basecoat and clear-coat depositions, from the examined samples, only TCP + CED and the HDG/TCP + CED coating systems are eligible processes.

**Author Contributions:** T.D. and Y.A.M. have prepared the coating samples and carried out EIS/polarization measurements. They have also carried out the SEM observations and XRD analyses and have also helped with the general organization and with the English written manuscript; W.R.O. has contributed to the general organization of the experimentations and analyses and correlations.

He has also written and organized the proposed manuscript. All authors have read and agreed to the published version of the manuscript.

**Funding:** Acknowledgments are also provided to PPG Brazil and to the financial support provided by FAEPEX-UNICAMP (#2407/21 and # 2120/21), CAPES (Coordination for the Improvement of Higher Education Personnel), Ministry of Education, Brazil, Grant #1) and CNPq (The Brazilian Research Council) Grants, #405602/2018-9 and #304950/2017-3.

**Institutional Review Board Statement:** Not applicable.

**Informed Consent Statement:** Not applicable.

**Data Availability Statement:** The authors also declare that all research data supporting this publication are directly available within this publication.

**Acknowledgments:** An anonymous colleague, who is a native speaker, has significantly contributed to English writing revision. The authors acknowledge this mentioned colleague.

**Conflicts of Interest:** The authors declare no conflict of interest.

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
