# Peer review of "The Holes of Zn Phosphate and Hot Dip Galvanizing on Electrochemical Behaviors of Multi-Coatings on Steel Substrates"

_metals, doi:10.3390/met12050863_

Round 1

Reviewer 1 Report

The manuscript entitled “The Holes of Zn Phosphate and Hot Dip Galvanizing on Electrochemical Behaviors of Multi-coatings on Steel Substrates" (Metals-1693687)  shows an interesting corrosive study of the distinctive coatings applied in the automotive industry. The authors present a detailed description of the EIS technique and potentiodynamic polarization curves, among other.  Also, the cost-to-efficiency to relative coating density ratio is evaluated. My suggestion is to accept the manuscript.

Author Response

Dear Reviewer, Authors are very grateful with revisions/comments provided.

Reviewer 2 Report

The article presents the results of corrosion resistance tests of a complex corrosion protection system consisting of tricationic phophate (TCP) conversion layer and painting layer obtained by cathodic electrodeposition (CED). Corrosion tests were carried out with various configurations of the protective system formed on the surface of a steel sheet or the surface of a previously galvanized steel sheet.

The tests were carried out on steel sheets, but they concerned coatings, one of which was a conversion coating, and the other a paint (organic) coating. Both the conversion coating and the epoxy coating (the paint coating is not sufficiently characterized in the article) are not metal.

I believe that both the research topic and the materials tested are more suitable for publication in “Coatings” or corrosion journals than in “Metals”.

Notes to the article:

Line 38: The authors focus in this article on anti-corrosive treatments designed to protect against electrochemical corrosion. In electrochemical corrosion processes, the corrosion products of steel are mainly iron hydroxides. In the process of chemical (high-temperature) corrosion, oxides, mainly FeO and Fe3O4, are formed. Fe2O3 oxide is formed at temperatures above 800oC and its share is very small. The statement in line 38 that corrosion leads to the formation of (only) Fe2O3 oxide on steel is misleading.

Line 85-86: I am not aware that Cr (III) coatings are forbidden. Rather, they represent an alternative to the genuinely forbidden Cr (VI) coatings.

Lines 198-202: When describing the salt spray test, indicate the pH of the solution and the temperature.

Figure 2: Graph descriptions are illegible.

Figure 7: On what basis do the authors claim that the corrosion product is FeOOH. The further part of the article presents XRD tests, but at this stage it is not known on what basis the scheme was made. It is also not known what causes the ferritic-pearlitic structure of the steel. In the description of the materials, the chemical composition is given (these are rather the requirements of the standard), but it is not known what type of steel it is.

Line 601: EDX was given while abbreviation EDS was given on line 247. Uniform nomenclature should be used.

Figure 12: The corrosion products of the zinc coating have not been identified.

Conclusion 1 and 2 do not result from the obtained test results, but are only an assessment of the possibilities of research methods that are commonly known.

Conclusion 4: "corrosion resistance is associated with adhesion of the CED layer on TCP coating" - On what basis was it found? The article does not present research on this topic.

Author Response

Dear Editor;

Please, the revised version of the proposed manuscript is attached. All Reviewers’ suggestions are provided and solved. A compilation of these modifications/improvements with each Reviewers’ comments are followed:

Best Regards;

Wislei Riuper Osório

University of Campinas, UNICAMP

Limeira, São Paulo, Brazil

_ _ _ __

Reviewer(s)' Comments to Author:

Reviewer: 2

“The article presents the results of corrosion resistance tests of a complex corrosion protection system consisting of tricationic phophate (TCP) conversion layer and painting layer obtained by cathodic electrodeposition (CED). Corrosion tests were carried out with various configurations of the protective system formed on the surface of a steel sheet or the surface of a previously galvanized steel sheet.

The tests were carried out on steel sheets, but they concerned coatings, one of which was a conversion coating, and the other a paint (organic) coating. Both the conversion coating and the epoxy coating (the paint coating is not sufficiently characterized in the article) are not metal.

I believe that both the research topic and the materials tested are more suitable for publication in “Coatings” or corrosion journals than in “Metals””.

AUTHORS: Authors are very grateful with comments and suggestions provided. Based on al Reviewers’ comments, we have decided to continue with submitting in Special Issue in Metals (MDPI)

“Notes to the article:

  • Line 38: The authors focus in this article on anti-corrosive treatments designed to protect against electrochemical corrosion. In electrochemical corrosion processes, the corrosion products of steel are mainly iron hydroxides. In the process of chemical (high-temperature) corrosion, oxides, mainly FeO and Fe3O4, are formed. Fe2O3 oxide is formed at temperatures above 800oC and its share is very small. The statement in line 38 that corrosion leads to the formation of (only) Fe2O3 oxide on steel is misleading.”

AUTHORS: The Reviewer is correct. Based ion this comment, Authors have provided modifications which are yellow highlighted, as follow:

“In the case of steel, it corrodes to many oxidated species, such as magnetite (Fe3O4) and iron oxide (FeO). These species have composition closely with the raw material that originates this alloy [2, 3].”

New reference was added:

[3] MORCILLO, M. et al. Environmental Conditions for Akaganeite Formation in Marine Atmosphere Mild Steel Corrosion Products and Its Characterization. CORROSION, [S. l.], v. 71, n. 7, p. 872–886, 2015 a. ISSN: 0010-9312, 1938-159X. DOI: 10.5006/1672.

  • “Line 85-86: I am not aware that Cr (III) coatings are forbidden. Rather, they represent an alternative to the genuinely forbidden Cr (VI) coatings.”

AUTHORS: The Reviewer is also correct. Authors are grateful with this observation. New sentence was included as yellow highlighted, as follow: “Particularly, those containing chromate [20] are toxic and/or carcinogenic, which have the tendency to been banned.”

  • “Lines 198-202: When describing the salt spray test, indicate the pH of the solution and the temperature.”

AUTHORS: The reviewer’s comment is very important. Authors have appreciated this suggestion. New sentence was included, as yellow highlighted:

“These were conducted according to the ASTM B 117 standard using a chamber (Model SST-B, Ten Billion Co., Taiwan) under an atmosphere containing 5 wt.% NaCl, solution pH of 7 and room temperature.”

  • “Figure 2: Graph descriptions are illegible.”

AUTHORS: Please, notice Figure 2 was changed to facilitate the interpretation.

  • “Figure 7: On what basis do the authors claim that the corrosion product is FeOOH. The further part of the article presents XRD tests, but at this stage it is not known on what basis the scheme was made. It is also not known what causes the ferritic-pearlitic structure of the steel. In the description of the materials, the chemical composition is given (these are rather the requirements of the standard), but it is not known what type of steel it is.”

AUTHORS: The presence of FeOOH is known for the corrosion process of steel substrates. Additionally, we found this compound in XRD in the substrate after the EIS experiment.

CASE OF FeOOH PRESENCE:

Please, new sentences were included between lines 456 and 462, as yellow highlighted:

Amirudin and Thierry [41] and other colleagues [18] have proposed a review paper explaining corrosion mechanism of phosphate zinc layer on steel substrate. With very high scientific merit they have explained the effects of various factors influencing the corrosion mechanism. These mentioned works help to understand various stages observed in this investigation. Although not represented in Figure 7, when only CED coating is considered, it can schematically be represented by Figure 7a. In addition, Figure 7 shows the presence of the FeOOH as corrosion by-product occurring into a CRS. Morcillo et al. [3] have detail the importance of the FeOOH and their species in the steel corrosion. Also, the presence of FeOOH was found in CRS samples available after EIS in the present work, as will be discussed in the next section”.

CASE OF FERRITIC-PEARLITIC STRUCTURE:

Please, notice that new sentence (into Figure’s caption) was also included, as yellow highlighted:

Figure 7: “Schematic representation with equivalent circuit proposed: (a) substrate and TCP coating, (b) substrate TCP + CED coating systems, and (c) complete/reorganized equivalent circuit. The resulting microstructures and microconstituents of a mild steel (similar to CRS) is agreed previous reported investigation [27].”

CASE OF CRS TYPE OF STEEL:

Between lines 161 and 165, the type of steel is indicated as CRS (cold rolled steel). Their corresponding composition is described and the standard composition certified by a supplier. So, the CRS is in the group of low carbon steel. New sentence was included, as yellow highlighted: “A cold rolled steel (CRS), belonging in the group of low carbon steel [27], as substrate was used.”

  • “Line 601: EDX was given while abbreviation EDS was given on line 247. Uniform nomenclature should be used.”

AUTHORS: Authors apologize for the inconvenience. Nomenclature was standardized in lines 250 and 585.

  • “Figure 12: The corrosion products of the zinc coating have not been identified.”

AUTHORS: The reviewer is correct. Authors thank for this observation. At line 802, it is explained the selection of general zinc complex formation. It was changed the intermediary in the Figure 12 as at line 802 (ZnXa(OH)y).

  • “Conclusion 1 and 2 do not result from the obtained test results, but are only an assessment of the possibilities of research methods that are commonly known.”

AUTHORS: The reviewer is correct. Based on this comment, Authors have deleted it.

  • “Conclusion 4: "corrosion resistance is associated with adhesion of the CED layer on TCP coating" - On what basis was it found? The article does not present research on this topic.”

AUTHORS: The reviewer is correct. Authors thank Reviewer due to this comment. The statement is widely cited in papers regarding zinc phosphate application and characteristics, as previously reported:

AMIRUDIN, A.; THIERRY, D. Corrosion mechanisms of phosphated zinc layers on steel as substrates for automotive coatings. Progress in Organic Coatings, [S. l.], v. 28, n. 1, p. 59–75, 1996. ISSN: 03009440. DOI: 10.1016/0300-9440(95)00554-4.

TEGEHALL, P. E.; VANNERBERG, N. G. Nucleation and formation of zinc phosphate conversion coating on cold-rolled steel. Corrosion Science, [S. l.], v. 32, n. 5–6, p. 635–652, 1991. ISSN: 0010938X. DOI: 10.1016/0010-938X(91)90112-3.

RANI, Nitu; SINGH, Arun K.; ALAM, Sarfaraz; BANDYOPADHYAY, N.; DENYS, M. B. Optimization of phosphate coating properties on steel sheet for superior paint performance. Journal of Coatings Technology and Research, [S. l.], v. 9, n. 5, p. 629–636, 2012. ISSN: 1547-0091, 1935-3804. DOI: 10.1007/s11998-012-9395-9.

Reviewer 3 Report

This manuscript evaluates four unique coating systems on steel substrates: TCP, CED, TCP+CED, and HDG+TCP+CED coatings. However, there has been a lot of research on holes of Zn phosphate and hot dip galvanizing. Especially, the Hot dip galvanizing process for steel corrosion substrate has been reported in many articles, such as Surface Topography: Metrology and Properties, 2021, 9(4): 045037; Materials Today: Proceedings, 2021, 46: 6700-6703; Corrosion Science, 2020, 174: 108846. Therefore, I don’t think the research involved in this article is innovative and the methods in this manuscript do not have obvious advances than those previously reports. So, I think that this manuscript is not appropriate for metals. Some of my observations are noted below:

  • Regarding the fitting of the polarization curve, check whether there is a Tafel area. Tafel fittings need to be done in the Tafel area.
  • The reference electrode type to which the potential is referred should be as shown in Figure 2.
  • For the EIS results, data points are missing in the low frequency region in Figures 3 and 4. Please confirm the authenticity and integrity of the data.
  • In the salt spray section, images of each sample before the salt spray test should be provided for comparison

Author Response

Reviewer: 3

  • “This manuscript evaluates four unique coating systems on steel substrates: TCP, CED, TCP+CED, and HDG+TCP+CED coatings. However, there has been a lot of research on holes of Zn phosphate and hot dip galvanizing. Especially, the Hot dip galvanizing process for steel corrosion substrate has been reported in many articles, such as Surface Topography: Metrology and Properties, 2021, 9(4): 045037; Materials Today: Proceedings, 2021, 46: 6700-6703; Corrosion Science, 2020, 174: 108846. Therefore, I don’t think the research involved in this article is innovative and the methods in this manuscript do not have obvious advances than those previously reports. So, I think that this manuscript is not appropriate for metals.”

AUTHORS: The Reviewer is correct. The suggested articles were consulted. Authors appreciate this comment. The fast corrosion evaluation of coated substrates using EIS is commonly reported. For this reason, it is believed that it is an innovative method to predict corrosion performance comparatively.

  • “Regarding the fitting of the polarization curve, check whether there is a Tafel area. Tafel fittings need to be done in the Tafel area.”

AUTHORS: The Reviewer is also correct. Authors have commonly used Tafel’s extrapolation to determine the corrosion current density. When considering the both anode and cathode branches, usually Tafel fitting is matching with its area. The follow reference has frequently used:

[XX] McCafferty E. Validation of corrosion rates measured by the Tafel extrapolation method. Corrosion Science 47 (2005) 3202–3215.

  • “The reference electrode type to which the potential is referred should be as shown in Figure 2.”

AUTHORS: Again, the Reviewer is also correct. Authors thank Reviewer with this comment. Please, new sentence was included, as yellow highlighted:

Figure 2: “Experimental potentiodynamic polarization curves of three distinct coating systems (TCP, CED and TCP +CED) in 0.5 M NaCl solution, platinum plate as counter electrode and SCE as reference electrode, where: (a) the samples have no incisions (damage-induced samples), and (b) the samples with incisions exposing the steel substrate.”

  • “For the EIS results, data points are missing in the low frequency region in Figures 3 and 4. Please confirm the authenticity and integrity of the data.”

AUTHORS: Authors thank Reviewer’s comment. Figs. 3 and 4 were revised.

  • “In the salt spray section, images of each sample before the salt spray test should be provided for comparison”

AUTHORS: Thank you for the suggestion. The images were added in Figure 13.

Round 2

Reviewer 2 Report

The authors took into account all my comments. They supplemented the text of the article and introduced the necessary references.

If the subject matter presented by the authors concerning non-metal coatings is not objectionable and falls within the scope of the Metals journal, the article may be published.

Reviewer 3 Report

The manuscript meets the journal requirements and is recommended for acceptance